# Rhythmic coordination of hippocampal neurons during associative memory processing

Lara M Rangel[1,2,3*†‡], Jon W Rueckemann[1†], Pamela D Riviere[1†],
Katherine R Keefe[1], Blake S Porter[1,4], Ian S Heimbuch[1,5], Carl H Budlong[1],
Howard Eichenbaum[1]

[1]Center for Memory and Brain, Boston University, Boston, United States; [2]Cognitive Rhythms Collaborative, Boston University, Boston, United States; [3]Bioengineering Department, University of California, San Diego, La Jolla, United States; [4]University of Otago, Dunedin, New Zealand; [5]University of California, Los Angeles, Los Angeles, United States

**Abstract** Hippocampal oscillations are dynamic, with unique oscillatory frequencies present during different behavioral states. To examine the extent to which these oscillations reflect neuron engagement in distinct local circuit processes that are important for memory, we recorded single cell and local field potential activity from the CA1 region of the hippocampus as rats performed a context-guided odor-reward association task. We found that theta (4–12 Hz), beta (15–35 Hz), low gamma (35–55 Hz), and high gamma (65–90 Hz) frequencies exhibited dynamic amplitude profiles as rats sampled odor cues. Interneurons and principal cells exhibited unique engagement in each of the four rhythmic circuits in a manner that related to successful performance of the task. Moreover, principal cells coherent to each rhythm differentially represented task dimensions. These results demonstrate that distinct processing states arise from the engagement of rhythmically identifiable circuits, which have unique roles in organizing task-relevant processing in the hippocampus.

**\*For correspondence:** lara.m. rangel@gmail.com

[†]These authors contributed equally to this work

**Present address:** [‡]Cognitive Science, University of California, San Diego, La Jolla, United States

## Introduction

Neural oscillations arise from the temporal coordination of activity in organized networks of neurons (*Buzsáki and Draguhn, 2004*). The unique connectivity of a network constrains the number of distinct rhythmic profiles that its local circuits can manifest, and the input to the network at a given time dictates the rhythmic circuits that are engaged (*Cannon et al., 2014*). The dynamics of rhythms can thus reflect fast-paced changes in the coordination of activity within local circuits during information processing. Changes in the oscillatory activity of the hippocampus, a brain structure important for memory function, occur as it processes information it receives from multiple brain regions (*Buzsáki and Draguhn, 2004*; *Cannon et al., 2014*; *Colgin et al., 2009*; *Schomburg, et al., 2014*; *Lee et al., 1994*; *Igarashi et al., 2014*). By studying the interactions of hippocampal neurons with their rhythmic circuits, we gain insight into how single neuron activity is coordinated into the local circuit and systems level processes that support memory. Although great advances have been made in describing both single cell and rhythmic correlates of memory in hippocampal circuits, relatively few studies examine the interaction of these phenomena.

The hippocampus exhibits a diversity of rhythms (*Cannon et al., 2014*; *Buzsáki, 2002*; *Buzsáki and Freeman, 2015*; *Colgin and Moser, 2010*). The theta (4–12 Hz) rhythm is a dominant rhythm in the hippocampus that engages both principal and interneuron cell types, and depends on inputs from the medial entorhinal cortex (MEC) and the medial septum (*Lee et al., 1994*;

**eLife digest** Electrodes placed on the surface of the scalp can reveal rhythmic patterns of electrical activity within the brain. These rhythms reflect the coordinated firing of large numbers of neurons that are connected together within a network in order to process information. A single network can show rhythms with various different frequencies depending on its local connections and the pattern of input that it receives at any given time.

One region that exhibits striking changes in these rhythmic patterns is the hippocampus: a brain area that plays a key role in memory. The hippocampus contains many cell types, including interneurons (which form connections with nearby cells) and principal cells (which connect with cells outside of this region). Though both participate in rhythmic circuits, little is known about the different extents to which these distinct cell types are engaged in rhythmic processing, or how rhythmic processing might support memory.

Rangel, Rueckemann, Rivière et al. have now addressed these questions by using electrodes to record from the hippocampus as rats learned to associate specific odors in different environments with a reward. As the rats sniffed the odors, their brains showed four different hippocampal rhythms: from a low frequency called "theta", through "beta" and "low gamma" up to "high gamma" frequencies. Each of these hippocampal rhythms varied in strength over time, indicating that rhythmic processing is dynamic during the task.

Rangel, Rueckemann, Rivière et al. found that neurons fired rhythmically during trials in which the rat chose the correct odor-environment combination. In these correct trials, individual principal cells were more likely to fire in synchrony with only one of the rhythms. In contrast, interneurons were more likely to fire in synchrony to each of the four rhythms at some point during a correct choice. Among the four rhythms, coordinated principal cell and interneuron firing with respect to the beta rhythm was most tightly linked with a correct choice. These findings reveal that investigation of rhythmic dynamics in the hippocampus can provide insight into how the timing of cell activity is coordinated to support memory.

Buzsáki, 2002; Kocsis et al., 1999; Montgomery et al., 2009; Kubie et al., 1990). The hippocampus also exhibits oscillations in the beta and gamma frequency ranges that span from 15-150Hz (Buzsáki and Freeman, 2015; Colgin and Moser, 2010; Kay and Freeman, 1998; Martin et al., 2007; Gourevitch et al., 2010; Buzsáki and Schomburg, 2015). Changes in the prominence of these higher frequency oscillations can reflect changes in input from converging afferents. Specifically, slow and fast gamma oscillations in the CA1 region of the hippocampus are thought to arise from the influence of CA3 and MEC inputs, respectively (Colgin et al., 2009; Buzsáki and Schomburg, 2015; Schomburg et al., 2014). In addition, an intermediate beta frequency range in CA1 has been hypothesized to reflect inputs from the lateral entorhinal cortex (LEC) (Igarashi et al., 2014). These higher frequency oscillations often occur concurrently with the $\text{theta}_{4-12\text{Hz}}$ rhythm, and previous studies suggest that coordination of cell activity within co-occurring rhythms produces nested levels of organization in the hippocampal network (Colgin et al., 2009; Harris et al., 2003; Mizuseki et al., 2009; Buzsáki, 2010). Thus, the diverse rhythmic states observed in the hippocampus can reflect the coordination of distinct information processing.

The rhythms in the hippocampus are governed by the neurons that constitute its circuits. The diffuse, local projections of the interneuron population place them in an ideal position to shape the rhythmic organization of the network in response to inputs received from a diverse array of afferents (Freund and Buzsáki, 1996; Sik et al., 1995). Interneurons in the CA1 region differ greatly according to their thresholds of excitability, the decay time of their inhibition, and the subcellular compartments where they preferentially target principal cells (Cannon et al., 2014; Royer et al., 2012; Roux et al., 2014; Roux and Buzsáki, 2015). This diversity enables the interneuron population to flexibly sculpt the oscillatory profile of the hippocampal network while simultaneously shaping principal cell activity (Freund and Buzsáki, 1996; Sik et al., 1995; Sik et al., 1997). As the hippocampus integrates dynamic input during behavior, the interneurons can flexibly engage the appropriate circuits, dictating how the hippocampus processes information. Thus, changes in oscillations can indicate that hippocampal circuits have undergone a shift in processing state.

Such shifts in processing state can be observed through distinctive rhythmic dynamics in the hippocampus as it processes information during memory tasks. Transient increases in the amplitude of higher frequency beta and low gamma activity can be observed during the presentation of conditioned stimuli, suggesting that the hippocampus undergoes a change in processing state (*Igarashi et al., 2014*; *Kay and Freeman, 1998*; *Gourevitch et al., 2010*; *Berke et al., 2008*; *Rangel et al., 2015*). In addition, cross-frequency coupling in the hippocampus develops while learning context-guided odor-reward associations (*Tort et al., 2009; 2010*), which occurs concurrently with the development of odor-place conjunctive encoding in hippocampal principal neurons (*Komorowski et al., 2009*). Since the hippocampus exhibits distinctive rhythmic states during memory tasks, and several of them are tied to the onset of learning, these changes in oscillatory profiles could reflect circuit level processes supporting memory function. However, it remains unknown how the rhythmicity of hippocampal circuits relates to the activity of the constitutive neurons during memory processing.

We investigated the extent to which rhythmic engagement of distinct cell types during a memory task could support the ability of the hippocampus to represent associations. In previous studies, it has been shown that single neurons in the CA3 and CA1 regions of the hippocampus develop activity that is selective for odors, odor port locations, and conjunctions of particular odors at specific locations (*odor-position* selectivity) (*Komorowski et al., 2009*). We designed a novel task to spatially and temporally isolate the sampling of an olfactory cue from its behavioral outcome during a context-guided odor-reward association task. We then performed *in vivo* recordings of single cell and local field potential activity in the CA1 region of the rat hippocampus to characterize the relationship between individual neurons and local circuit dynamics.

We observed changes in theta (4–12 Hz), beta (15–35 Hz), low gamma (35–55 Hz), and high gamma (65–90 Hz) frequency power during odor sampling epochs when task-relevant information must be integrated for successful performance. $Theta_{4-12Hz}$, $beta_{15-35Hz}$, low $gamma_{35-55Hz}$, and high $gamma_{65-90Hz}$ rhythms differentially recruited principal cells and interneurons during successful performance of the task, suggesting that the different frequency bands represent functionally distinct processing states. Notably, principal cell and interneuron entrainment to $beta_{15-35Hz}$ frequency oscillations were the most correlated with correct performance. We propose that the $beta_{15-35Hz}$ rhythm instigates a processing of information in the hippocampus that is distinct from the processing that occurs in $theta_{4-12Hz}$, low $gamma_{35-55Hz}$, and high $gamma_{65-90Hz}$ and that the presence of the $beta_{15-35Hz}$ rhythm signals a recruitment of cell activity that may be critical for memory function.

## Results

We recorded both single cell and local field potential activity in the CA1 region of the dorsal hippocampus in order to determine their relationship during intervals when cues must be associated with a reward outcome. In our task, rats learned that pairs of odors have differential value (rewarded or unrewarded) depending upon the spatial context in which they are presented (*Figure 1a* (*top*), see Materials and methods). Rats first entered one of the two contexts and then sampled odors presented at two adjacent odor ports. The initiation of a poke triggered the release of an odor after a 250 ms delay. We analyzed neural activity during trials when the rat maintained a nose poke for 1.5 s while sampling a rewarded odor (correct trials) and during trials when the rat maintained a nose poke for 1.5 s while sampling the non-rewarded odor (incorrect trials). We recorded a total of 1368 cells (1301 principal cells, 67 interneurons) from 6 rats across a total of 45 sessions. Two half-sessions were recorded each day, and each half-session was analyzed separately (see Materials and methods).

### Dynamic rhythmic activity during the nose poke interval

We observed dynamic rhythmic activity during the nose poke interval. Prominent changes in amplitude were observed in the theta (4–12 Hz), beta (15–35 Hz), low gamma (35–55 Hz), and high gamma (65–90 Hz) frequency ranges (*Figure 1a* (*middle, bottom*), b-c). For each frequency band, we determined whether amplitude changed over the course of the nose poke or differed according to behavioral outcome (correct or incorrect). We performed a two-factor repeated measures ANOVA and found a significant main effect of time during the nose poke for all frequencies (*Figure 1e–h*; *time*: repeated measures $ANOVA_{theta}$: d.f. = 5, F= 10.32, p<0.00001; repeated measures

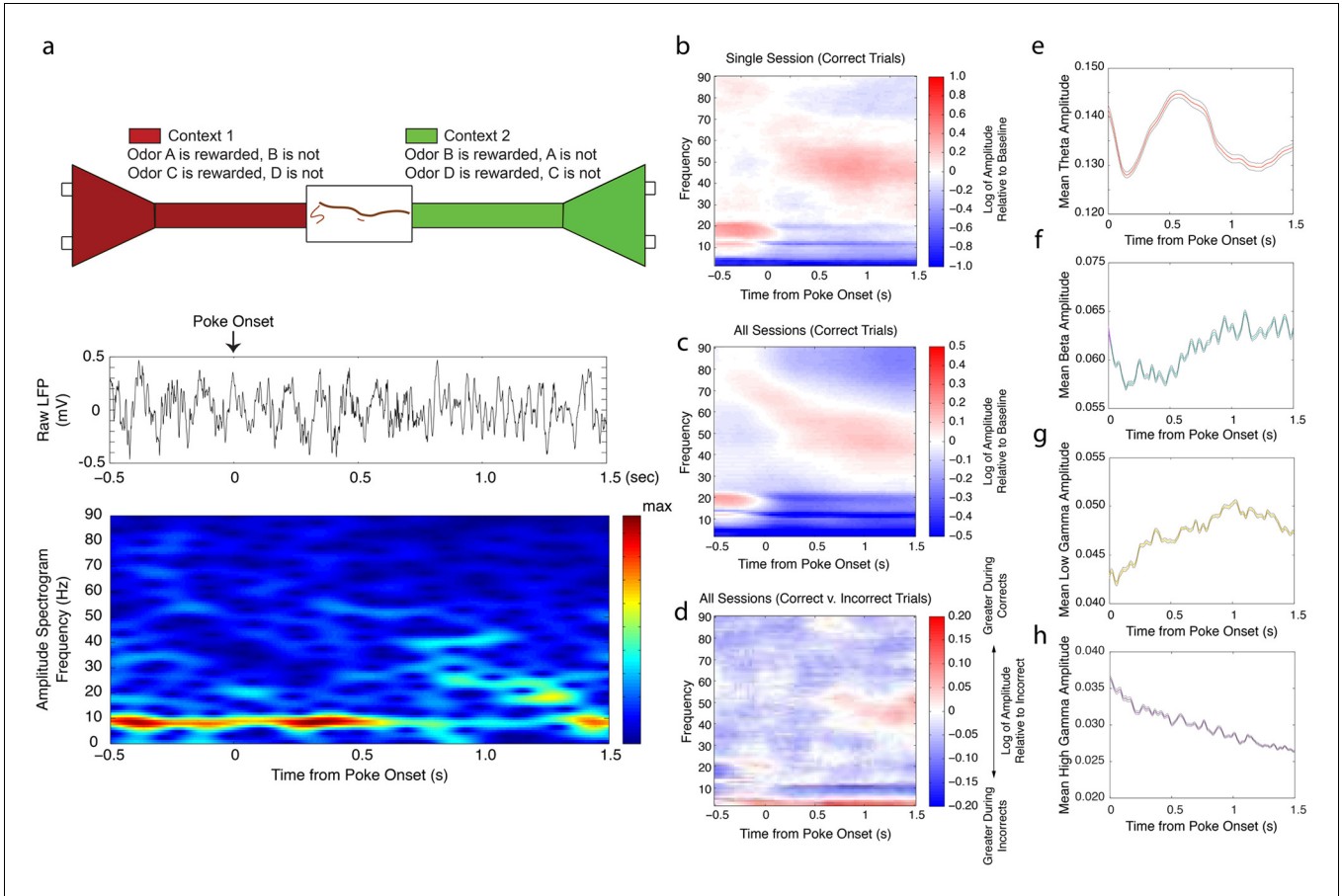

**Figure 1.** Changes in theta (4–12 Hz), beta (15–35 Hz), low gamma (35–55 Hz), and high gamma (65–90 Hz) amplitude during odor sampling intervals. (**a**) Schematic of our behavioral paradigm in which pairs of odors (odors A and B, and odors C and D are presented in blocks) are differentially rewarded depending upon the context in which they are presented (*top*), raw local field potential (LFP) trace (*middle*) and corresponding amplitude spectrogram (*bottom*) beginning 0.5 s prior to the initiation of a nose poke until 1.5 s after poke onset for a single correct trial. For a more detailed view of the automated apparatus, see *Figure 1—figure supplement 1*. (**b**) Amplitude spectrogram averaged across all correct trials for a single session, shown as the log of the amplitude relative to baseline inter-trial intervals. (**c**) Same as in **b**, averaged across all sessions. (**d**) Same as in **c**, but shown instead as the log of the amplitude of correct trials relative to incorrect trials. Low gamma$_{35-55Hz}$ amplitude demonstrates a greater increase over time during correct trials than incorrect trials. (**e-h**) Instantaneous amplitude of theta$_{4-12Hz}$ (**e**), beta$_{15-35Hz}$ (**f**), low gamma$_{35-55Hz}$ (**g**), and high gamma$_{65-90Hz}$ (**h**) during the 1.5 s odor-sampling interval.

The following figure supplement is available for figure 1:

**Figure supplement 1.** Automated behavioral apparatus.

ANOVA$_{beta}$: d.f. = 5, F= 23.87, p<0.00001; repeated measures ANOVA$_{low\ gamma}$: d.f. = 5, F= 17.34, p<0.00001; repeated measures ANOVA$_{high\ gamma}$: d.f. = 5, F= 63.78, p<0.00001), and no main effect for outcome (correct or incorrect) in any frequency (*outcome*: repeated measures ANOVA$_{theta}$: d.f. = 1, F= 1.19, p=0.2797, n.s.; repeated measures ANOVA$_{beta}$: d.f. = 1, F ≈ 0, p=0.9746, n.s.; repeated measures ANOVA$_{low\ gamma}$: d.f. = 1, F = 1.32, p=0.2546, n.s.; repeated measures ANOVA$_{high\ gamma}$: d.f. = 1, F = 0.08, p=0.7747, n.s.). These results indicate that while all four frequencies demonstrated significant changes in amplitude over the course of the nose poke, mean amplitudes were not significantly different across correct and incorrect trial types. However, we observed a significant interaction effect in the low gamma$_{35-55Hz}$ frequency range, due to increased low gamma$_{35-55Hz}$ amplitude during correct trials during the last second of the odor-sampling interval (time x outcome: repeated measures ANOVA$_{theta}$: d.f. = 5, F= 0.34, p=0.8886, n.s.; repeated measures ANOVA$_{beta}$: d.f. = 5, F = 1.46, p=0.2008, n.s.; repeated measures ANOVA$_{low\ gamma}$: d.f. = 5, F = 4.32, p=0.0008; repeated measures ANOVA$_{high\ gamma}$: d.f. = 5, F = 0.40, p=0.8513, n.s.). This increase in low gamma$_{35-55Hz}$

amplitude at the end of the nose poke during Correct Trials Only is evident in the ratio of the spectrograms for correct and incorrect trials (*Figure 1d*). This indicates that there is a change in processing over the course of the nose poke within low gamma$_{35-55Hz}$ rhythmic circuits that differentiates between correct and incorrect trials. Together, these results indicate that the nose poke interval contains a shift in processing state in the hippocampus, which is observable through the onset of changes in rhythmic circuits.

## Interneuron spike-phase coherence relationships to task performance

Populations of interneurons exhibited strong spike-phase coherence to the rhythms present during odor sampling. To test whether single cell entrainment to theta$_{4-12Hz}$, beta$_{15-35Hz}$, low gamma$_{35-55Hz}$, or high gamma$_{65-90Hz}$ frequency ranges was related to successful performance of the associative memory task, we first examined whether interneuron spike-phase coherence to each frequency range during the odor sampling interval was selective to correct or incorrect trial types. This interval was initiated by a nose poke, and continued as the poke was sustained for 1.5 s, when the rat committed to a decision. The interneurons (N = 67, 45 sessions with each half-session analyzed separately, 6 rats, see Materials and methods) were categorized as exhibiting significant spike-phase coherence to a given frequency range during Correct Trials Only, Incorrect Trials Only, or All (both correct and incorrect) Trials (*Figure 2a*). If single cell engagement in a rhythm in the form of spike-phase coherence is important for successful processing during the task, one might expect a larger number of cells to be coherent during Correct Trials Only. The converse might be true if single cell spike-phase coherence was to interfere with successful performance, resulting in a larger number of cells exhibiting significant spike-phase coherence during Incorrect Trials Only. Lastly, cells that exhibit significant spike-phase coherence to a rhythm on All Trials (correct and incorrect) might instead be engaged in underlying processes that are not task-specific. For each rhythm, the proportions of interneurons in each category were compared to the proportions that would be expected if the cells were equally distributed across all three categories. This comparison thus asks whether the number of cells exhibiting significant spike phase coherence to a rhythm is different across the three performance categories.

Of the interneurons that exhibited significant spike-phase coherence to beta$_{15-35Hz}$ (*Figure 2a*, *middle left*), the number that exhibited coherence to beta$_{15-35Hz}$ during Correct Trials Only was greater than the numbers coherent during Incorrect Trials Only or All Trials ($\chi^2_{beta}$ (2, N=66) = 51.54, p<0.00001; post hoc pairwise comparisons with Bonferroni adjusted alpha: $\chi^2_{correct\ v\ incorrect}$ (1, N=53) = 38.21, p<0.00001; $\chi^2_{correct\ v\ all}$ (1, N=62) = 20.90, p<0.00001; $\chi^2_{incorrect\ v\ all}$ (1, N=17) = 4.77, p=0.029, n.s.). Similarly, the number of interneurons coherent to high gamma$_{65-90Hz}$ (*Figure 2a*, *far right*) during Correct Trials Only was greater than the numbers coherent during Incorrect Trials Only or All Trials, with a larger number of interneurons coherent during All Trials than during Incorrect Trials Only as well ($\chi^2_{high\ gamma}$ (2, N=107) = 59.23, p<0.00001), post hoc pairwise comparisons with Bonferroni adjusted alpha: $\chi^2_{correct\ v\ incorrect}$ (1, N=71) = 59.51, p<0.00001; $\chi^2_{correct\ v\ all}$ (1, N=104) = 9.85, p=0.00017; $\chi^2_{incorrect\ v\ all}$ (1, N=39) = 27.92, p<0.00001). In contrast, the largest number of theta$_{4-12Hz}$ coherent interneurons (*Figure 2a*, *far left*) were coherent during All Trials, although a greater number of cells still exhibited coherence during Correct Trials Only compared to Incorrect Trials Only ($\chi^2_{theta}$ (2, N=126) = 80.19, p<0.00001, post hoc pairwise comparisons with Bonferroni adjusted alpha: $\chi^2_{correct\ v\ incorrect}$ (1, N=42) = 34.38, p<0.00001; $\chi^2_{correct\ v\ all}$ (1, N=124) = 15.61, p=0.00007; $\chi^2_{incorrect\ v\ all}$ (1, N=86) = 78.19, p<0.00001). Lastly, the numbers of interneurons coherent to low gamma$_{35-55Hz}$ (*Figure 2a*, *middle right*) during Correct Trials Only and All Trials were greater than the number coherent during Incorrect Trials Only, but were not significantly different from each other ($\chi^2_{low\ gamma}$ (2, N=91) = 37.21, p<0.00001), post hoc pairwise comparisons with Bonferroni adjusted alpha: $\chi^2_{correct\ v\ incorrect}$ (1, N=49) = 37.74, p<0.00001; $\chi^2_{correct\ v\ all}$ (1, N=88) = 0.18, p=0.6697, n.s.; $\chi^2_{incorrect\ v\ all}$ (1, N=45) = 33.80, p<0.00001). In summary, while the proportion of interneurons exhibiting coherence during Correct Trials Only or All Trials varies across each of the four rhythms, coherence exclusively during incorrect trials is quite rare. Moreover, the heterogeneity across rhythms indicates that each rhythmic circuit uniquely engages interneurons in processing states that differentially contribute to task performance.

To determine whether any of the rhythms are unique in their ability to engage interneuron activity during specific trial types, we also compared the distribution of interneurons across the three performance categories for all rhythms. The interneurons coherent to theta$_{4-12Hz}$ were distributed

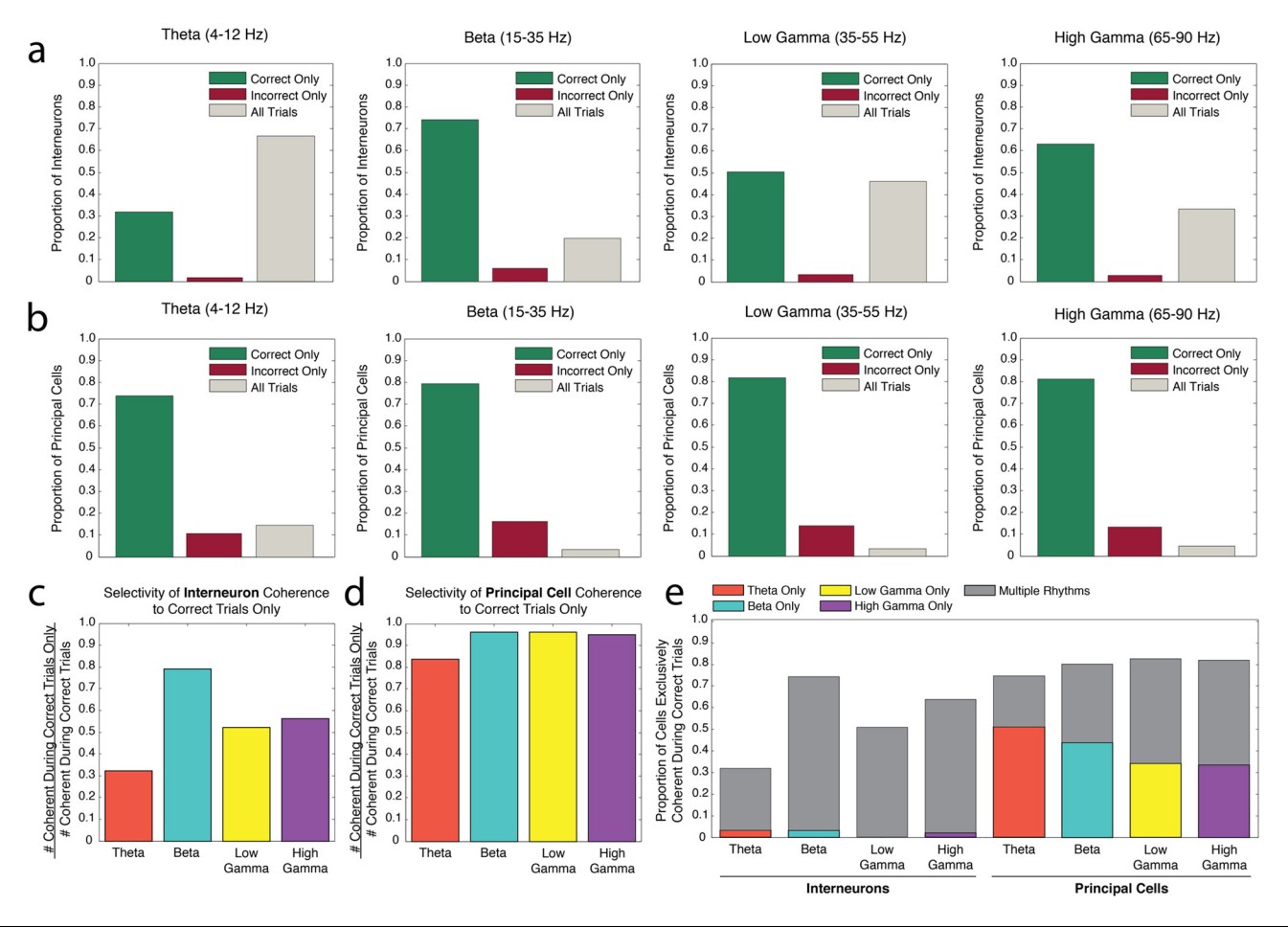

**Figure 2.** Interneuron and principal cell engagement in rhythmic circuits is related to task performance. (a) Proportions of interneurons demonstrating significant spike-phase coherence to theta$_{4-12Hz}$ (*far left*), beta$_{15-35Hz}$ (*middle left*), low gamma$_{35-55Hz}$ (*middle right*), and high gamma$_{65-90Hz}$ (*far right*) during Correct Trials Only (*green*), Incorrect Trials Only (*red*), or All Trials (*gray*). The largest proportion of theta$_{4-12Hz}$ coherent interneurons (*far left*) was coherent during All Trials, regardless of outcome. In contrast, the largest proportion of beta$_{15-35Hz}$ coherent interneurons (*middle left*) was coherent selectively during Correct Trials Only. (b) Same as in a, for the principal cell population. For each rhythm, the largest proportions of principal cells were coherent during Correct Trials Only. (c) The number of interneurons coherent during Correct Trials Only as a ratio of the total number of interneurons coherent during correct trials (# coherent during Correct Trials Only + # coherent during All Trials). (d) Same as in c, for the principal cell population. (e) The proportions of interneurons and principal cells coherent to each rhythm during Correct Trials Only, subdivided into the proportions exhibiting coherence to a single rhythm or multiple rhythms. While the interneuron population demonstrates flexible engagement into multiple rhythmic circuits during successful performance, principal cells are more often engaged in single rhythmic circuits.

The following source data is available for figure 2:

**Source data 1.** The number of interneurons within each rhythmic category that were coherent to each possible combination of the four rhythms.

**Source data 2.** The number of principal cells within each rhythmic category that were coherent to each possible combination of the four rhythms.

differently across the three performance categories than the interneurons coherent to beta$_{15-35Hz}$, low gamma$_{35-55Hz}$, or high gamma$_{65-90Hz}$ ($\chi^2_{theta-beta}$ (2, N=192) = 38.56, p<0.00001; $\chi^2_{theta-low\ gamma}$ (2, 217)= 9.21, p=0.009; $\chi^2_{theta-high\ gamma}$ (2, N=233) = 25.28, d.f. = 2, p<0.00001). Post hoc pairwise comparisons revealed that these differences were driven by the relative proportions of interneurons in the Correct Trials Only and All Trials categories, while similar proportions were observed in the Incorrect Trials Only category across rhythms (*theta-beta*: $\chi^2_{correct}$ (1, N=192) = 31.46, p<0.00001, $\chi^2_{incorrect}$ (1, N=192) = 2.86, p=0.0906, n.s., $\chi^2_{all}$ (1, N=192) = 38.23, p<0.00001; *theta-low gamma*: $\chi^2_{correct}$ (1, N=217) = 7.81, p=0.0052, $\chi^2_{incorrect}$ (1, N=217) = 0.69, p=0.4075, n.s., $\chi^2_{all}$ (1, N=217) =

9.13, p=0.0025; *theta-high gamma*: $\chi^2_{correct}$ (1, N=233) = 23.54, p<0.00001, $\chi^2_{incorrect}$ (1, N=233) = 0.41, p=0.5230, n.s., $\chi^2_{all}$ (1, N=233) = 25.26, p<0.00001; Bonferroni adjusted alpha). Thus, interneuron coherence during All Trials occurs more often in the $theta_{4-12H}$ rhythm, distinguishing it from other rhythms. In addition, interneurons coherent to low $gamma_{35-55Hz}$ were distributed differently across the three performance categories than the interneurons coherent to $beta_{15-35Hz}$ ($\chi^2_{beta-low\ gamma}$ (2, N=157) = 11.85, p=0.003), due to a greater degree of selectivity in the $beta_{15-35Hz}$ coherent population for engagement during Correct Trials Only ($\chi^2_{correct}$ (1, N=157) = 8.99, p=0.003, $\chi^2_{incorrect}$ (1, N=157) = 0.69, p=0.4075, n.s., $\chi^2_{all}$ (1, N=157) = 11.77, p=0.0006; Bonferroni adjusted alpha). The distributions across the three performance categories were not significantly different between $beta_{15-35Hz}$ and high $gamma_{65-90Hz}$ coherent interneurons ($\chi^2_{beta-high\ gamma}$ (2, N=173) = 4.56, p=0.1021, n.s.) or between low $gamma_{35-55Hz}$ and high $gamma_{65-90Hz}$ coherent interneurons ($\chi^2_{low\ gamma-high\ gamma}$ (2, N=198) = 3.44, d.f. = 2, p=0.1793, n.s.). To better illustrate differences observed across rhythms (*Figure 2c*), we plotted the ratio of the number of interneurons coherent during Correct Trials Only to the total number that exhibited coherence during correct trials (the combined Correct Trials Only and All Trials categories). These results indicate that interneuron engagement in certain rhythms can be differentially dependent upon task performance.

Notably, for each of the four rhythms, the smallest number of interneurons exhibited significant spike-phase coherence during Incorrect Trials Only. This decrease in interneuron spike-phase coherence during incorrect trials can also be observed by comparing the magnitude of coherence for the interneurons during correct and incorrect trials. Adjusting for firing rate differences between trial types (*Figure 3a, b*, see Materials and methods), we observed significant decreases in the strength of interneuron spike-phase coherence to each rhythm during incorrect trials when compared to correct trials (Median (Mdn)$_{theta-correct}$ = 0.1884, Mdn$_{theta-incorrect}$ = 0.0998, Wilcoxon signed-rank test Z = 4.76, p<0.00001; Mdn$_{beta-correct}$ = 0.0883, Mdn$_{beta-incorrect}$ = 0.0223, Wilcoxon signed-rank test Z = 9.25, p<0.00001; Mdn$_{low\ gamma-correct}$ = 0.0906 Mdn$_{low\ gamma-incorrect}$ = 0.0317, Wilcoxon signed-rank test Z = 7.27, p<0.00001; Mdn$_{high\ gamma-correct}$ = 0.0830 Mdn$_{high\ gamma-incorrect}$ = 0.0221, Wilcoxon signed-rank test Z = 8.28, p<0.00001). Since higher firing rates in phase-modulated cells can increase estimates of spike-phase coherence strength, we determined whether firing rate differences between correct and incorrect trials could explain the differences in selective coherence. If the decrease in coherence during Incorrect Trials Only is due to lower firing rates during incorrect trials, then we would observe significantly lower firing rates during incorrect trials compared to correct trials. To the contrary, we observed that interneurons exhibited significantly higher firing rates during incorrect trials than correct trials (*Figure 3c*; Mdn$_{correct}$ = 12.78 Hz, Mdn$_{incorrect}$ = 14.20 Hz, Wilcoxon signed-rank test Z = -3.22, p=0.0013). Thus, the lack of interneuron engagement during Incorrect Trials Only is not due to firing rate differences between correct and incorrect trial types. Instead, unsuccessful processing during the task coincides with a unique inability to engage the interneuron population in rhythmic circuits. Taken together, these results suggest that trial outcome is strongly related to interneuron engagement in each of the four rhythms.

To examine whether performance dependent engagement of the interneurons coincides with a rhythmic phase preference, we compared their average phase of spiking during correct and incorrect trial types (*Figure 3—figure supplement 1*, see Materials and methods). If engagement in a rhythm during Correct Trials Only represents participation in a rhythmic processing state that occurs uniquely during successful task performance, then the phase of interneuron spiking activity may change between correct and incorrect trials. In addition, interneurons exhibiting coherence to a rhythm during All Trials might exhibit the same phase preference during correct and incorrect trials because their participation is not related to successful task performance. We first tested the interneurons that exhibited significant spike-phase coherence to a specific rhythm during Correct Trials Only by performing circular correlations on their preferred (average) phase during correct and incorrect trials. Interneurons that were coherent to $theta_{4-12Hz}$, $beta_{15-35Hz}$, and high $gamma_{65-90Hz}$ during Correct Trials Only did not exhibit a consistent phase preference between correct and incorrect trials (*Figure 3—figure supplement 1c*; R$_{theta-correct}$ = 0.26, p=0.0823; R$_{beta-correct}$ = 0.12, p=0.4003; R$_{high\ gamma-correct}$ = 0.16, p=0.2012). Interneurons that were coherent to low $gamma_{35-55Hz}$ during Correct Trials Only exhibited only a weak correlation in phase preference between correct and incorrect trials (R$_{low\ gamma-correct}$ = 0.31, p=0.0258). Overall, the interneurons that were coherent during Correct Trials Only did not exhibit similar engagement across trial types as measured by spike-phase coherence. Given that the magnitude of coherence is greater across this population during correct trials,

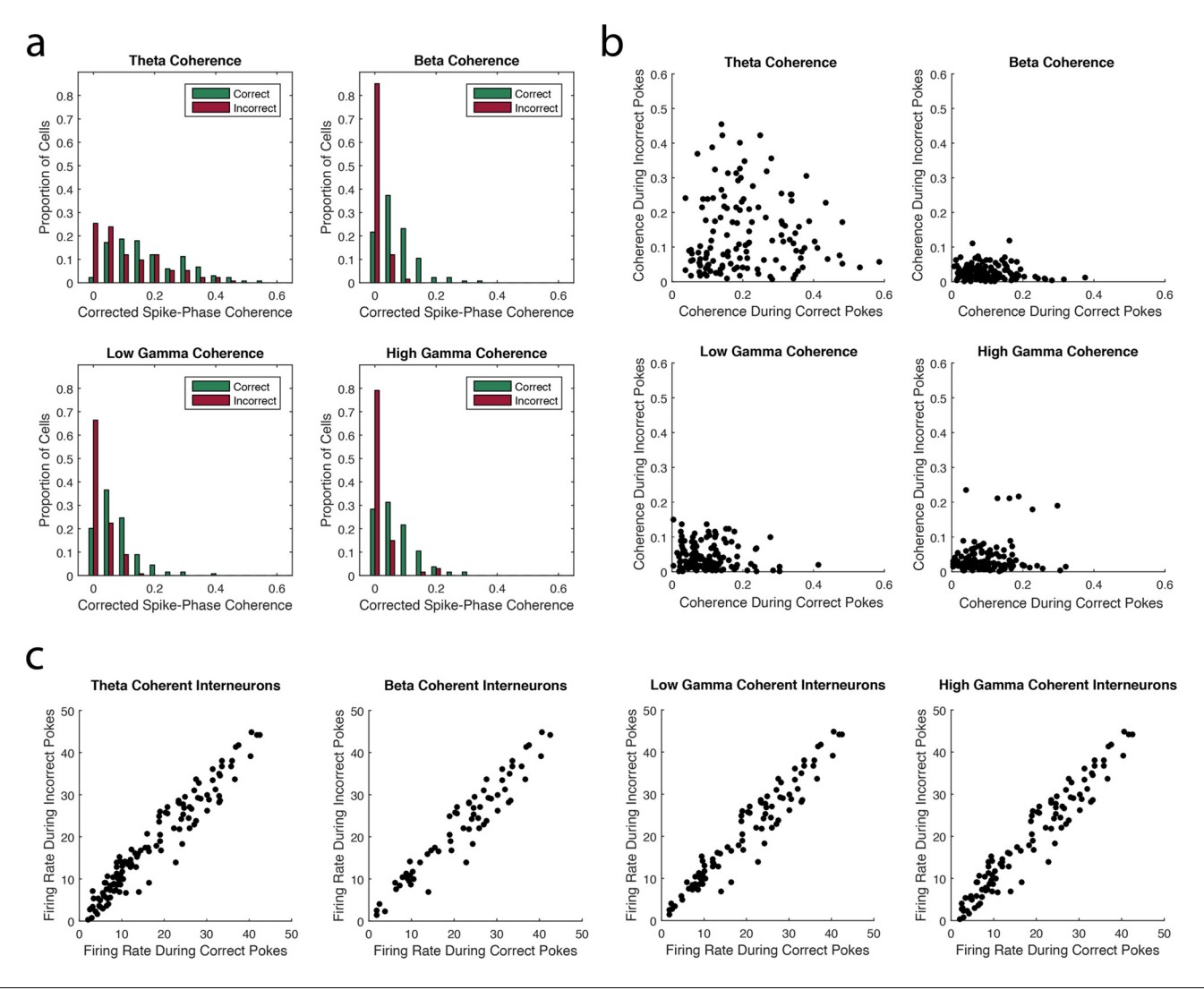

**Figure 3.** The strength of interneuron coherence to each rhythm is greater during correct trials than incorrect trials. (**a**) The proportions of interneurons exhibiting a given magnitude of coherence on the x-axis during correct (*green*) and incorrect (*red*) trials with respect to the theta$_{4-12Hz}$, beta$_{15-35Hz}$, low gamma$_{35-55Hz}$, or high gamma$_{65-90Hz}$ rhythm. Greater proportions of interneurons exhibit larger magnitudes of coherence during correct trials compared to incorrect trials. (**b**) The magnitude of coherence during correct trials plotted against the magnitude of coherence during incorrect trials for all interneurons that were coherent to each rhythm during either correct or incorrect trials. (**c**) The average firing rate during correct trials plotted against the average firing rate during incorrect trials for all interneurons that were coherent to each rhythm during either correct or incorrect trials.

The following figure supplement is available for figure 3:

**Figure supplement 1.** The phase of interneuron coherence to each rhythm during correct and incorrect trials.

together these results suggest that there is a critical reorganization of spike timing in these interneurons during successful processing in the hippocampus. In stark contrast, interneurons that were coherent to theta$_{4-12Hz}$, beta$_{15-35Hz}$, low gamma$_{35-55Hz}$, and high gamma$_{65-90Hz}$ during All Trials exhibited relatively strong correlations between the average phases observed across the population during correct and incorrect trials (*Figure 3—figure supplement 1d*; $R_{theta-all}$ = 0.82, p<0.00001; $R_{beta-all}$ = 0.55, p=0.0402; $R_{low\ gamma-all}$ = 0.41, p=0.0131; $R_{high\ gamma-all}$ = 0.66, p=0.0002). Thus the interneurons that are coherent during All Trials also have similar phases of entrainment during correct and incorrect trials, suggesting that they are similarly engaged in processing within local rhythmic circuits irrespective of whether the rat successfully performs the task. Combined with evidence

that each rhythm demonstrates a unique ability to engage interneurons across correct and incorrect trials (*Figure 2a*), these results demonstrate that each rhythmic circuit differentially participates in task-related processing as demonstrated by the selective entrainment of interneuron spike timing.

## Principal cell spike-phase coherence relationships to task performance

Spike-phase coherence analyses were then performed on the principal cell population. All principal cells (N = 1301, 45 sessions with each half-session analyzed separately, 6 rats) were categorized as exhibiting significant spike-phase coherence to a given frequency range during Correct Trials Only, Incorrect Trials Only, or All Trials (*Figure 2b*). For all rhythms, principal cells preferentially exhibited significant spike-phase coherence during Correct Trials Only (*theta*: $\chi^2_{theta}$ (2, N=349) = 266.86, p<0.00001, post hoc pairwise comparisons with Bonferroni adjusted alpha: $\chi^2_{correct\ v\ incorrect}$ (1, N=298) = 165.38, p<0.00001; $\chi^2_{correct\ v\ all}$ (1, N=311) = 140.45, p<0.00001; $\chi^2_{incorrect\ v\ all}$ (1, N=89) = 1.90, p=0.1682, n.s.; *beta*: $\chi^2_{beta}$ (2, N=91) = 92.40, p<0.00001, post hoc pairwise comparisons with Bonferroni adjusted alpha: $\chi^2_{correct\ v\ incorrect}$ (1, N=88) =38.23, p<0.00001; $\chi^2_{correct\ v\ all}$ (1, N=76) = 64.47, p<0.00001; $\chi^2_{incorrect\ v\ all}$ (1, N=18) = 8.00, p=0.0047; *low gamma*: $\chi^2_{low\ gamma}$ (2, N=120) = 132.65, p<0.00001, post hoc pairwise comparisons with Bonferroni adjusted alpha: $\chi^2_{correct\ v\ incorrect}$ (1, N=116) = 57.97, p<0.00001; $\chi^2_{correct\ v\ all}$ (1, N=101) = 85.63, p<0.00001; $\chi^2_{incorrect\ v\ all}$ (1, N=21) = 8.05, p=0.0045; *high gamma*: $\chi^2_{high\ gamma}$ (2, N=134) = 132.65, p<0.00001, post hoc pairwise comparisons with Bonferroni adjusted alpha: $\chi^2_{correct\ v\ incorrect}$ (1, N=128) = 66.13, p<0.00001; $\chi^2_{correct\ v\ all}$ (1, N=116) = 93.24, p<0.00001; $\chi^2_{incorrect\ v\ all}$ (1, N=24) = 6.00, p=0.0143). In addition, for every rhythm except theta$_{4-12Hz}$ (*Figure 2b, far left*), the number of principal cells coherent during incorrect trials was significantly greater than the number coherent during All Trials. Since the entrainment of principal cells in rhythmic circuits occurs most often during Correct Trials Only, these data suggest that principal cell engagement is a unique feature of successful processing in the hippocampus.

To determine whether any of the rhythms are unique in their ability to engage principal cell activity during specific trial types, we also compared the distributions of principal cells across the three performance categories for all pairs of rhythms. The distributions of principal cells across the three performance categories were not significantly different in beta$_{15-35Hz}$, low gamma$_{35-55Hz}$, or high gamma$_{65-90Hz}$ coherent principal cells ($\chi^2_{beta-low\ gamma}$ (2, N=211) = 0.216, p=0.8976, n.s.; $\chi^2_{beta-high\ gamma}$ (2, N=225) = 0.556, p=0.7573, n.s.; $\chi^2_{low\ gamma-high\ gamma}$ (2, N=254) = 0.237, p=0.8883, n.s.). However, the distribution of theta$_{4-12Hz}$ coherent principal cells was significantly different than the distributions in all other rhythms ($\chi^2_{theta-beta}$ (2, N=440) = 9.72, p=0.0078; $\chi^2_{theta-low\ gamma}$ (2, N=469) = 11.25, p=0.0035; $\chi^2_{theta-high\ gamma}$ (2, N=483) = 9.69, p=0.0078). Post hoc pairwise comparisons revealed that this is due to a larger number of theta coherent principal cells that were coherent during All Trials (*theta-beta*: $\chi^2_{correct}$ (1, N=440) = 1.28, p=0.2573, n.s., $\chi^2_{incorrect}$ (1, N=440) = 2.13, p=0.1442, n.s., $\chi^2_{all}$ (1, N=440) = 8.59, p=0.0033; *theta-low gamma*: $\chi^2_{correct}$ (1, N=469) = 3.18, p=0.0744, n.s., $\chi^2_{incorrect}$ (1, N=467) = 0.93, p=0.3356, n.s., $\chi^2_{all}$ (1, N=469) = 10.98, p=0.0009; *theta-high gamma*: $\chi^2_{correct}$ (1, N=483) = 3.11, p=0.0777, n.s., $\chi^2_{incorrect}$ (1, N=483) = 0.61, p=0.4340, n.s., $\chi^2_{all}$ (1, N=483) = 9.56, p=0.0020). To better illustrate this difference in the theta$_{4-12Hz}$ coherent population compared to other rhythms (*Figure 2d*), we plotted the ratio of principal cells coherent during Correct Trials Only to the total number coherent during correct trials (the combined Correct Trials Only and All Trials categories). We demonstrate that principal cells exhibiting coherence to correct trials are almost exclusively coherent to Correct Trials Only for beta$_{15-35Hz}$, low gamma$_{35-55Hz}$, and high gamma$_{65-90Hz}$, while the theta$_{4-12Hz}$ coherent population exhibits a significantly greater number of neurons that are coherent to All Trials. In summary, these results suggest that rhythmic entrainment of principal neurons to beta$_{15-35Hz}$, low gamma$_{35-55Hz}$, and high gamma$_{65-90Hz}$ is closely related to processing that is specific to the memory task, while theta$_{4-12Hz}$ rhythmic entrainment serves a less performance-specific function.

Similar to the interneuron population, the smallest number of principal cells exhibited significant spike-phase coherence during Incorrect Trials Only in each of the four rhythms. This decrease in principal cell spike-phase coherence during incorrect trials was also observed through significant decreases in the strength of principal cell spike-phase coherence to each rhythm during incorrect trials when compared to correct trials, adjusted for firing rate differences between trial types (*Figure 4a and b*, Median (Mdn)$_{theta-correct}$ = 0.0499, Mdn$_{theta-incorrect}$ = 0.0356, Wilcoxon signed-rank test Z = 6.69, p<0.00001; Mdn$_{beta-correct}$ = 0.0351, Mdn$_{beta-incorrect}$ = 0.0172, Wilcoxon signed-

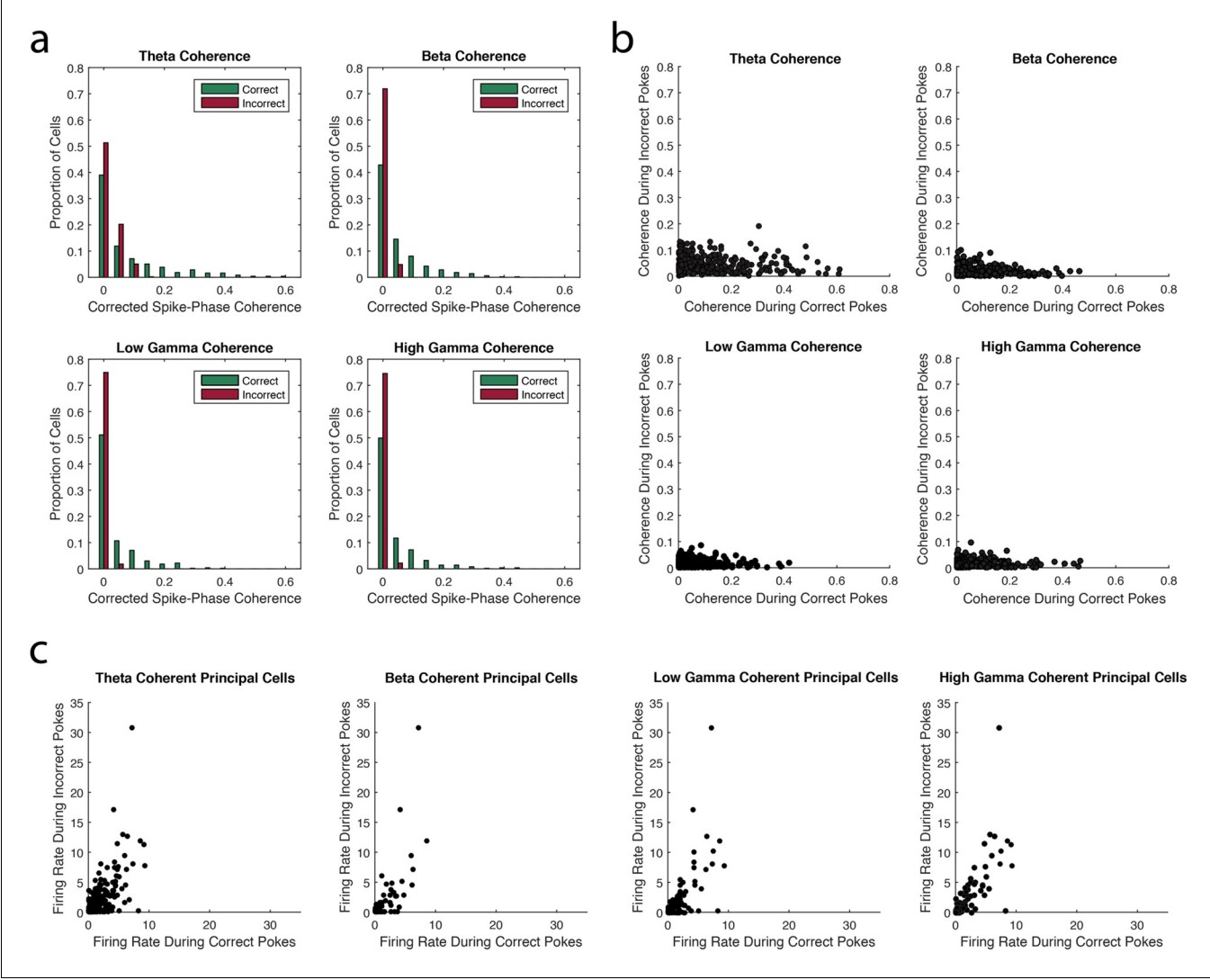

**Figure 4.** The strength of principal cell coherence to each rhythm is greater during correct trials than incorrect trials. (**a**) The proportion of principal cells exhibiting a given magnitude of coherence to theta$_{4-12Hz}$, beta$_{15-35Hz}$, low gamma$_{35-55Hz}$, or high gamma$_{65-90Hz}$ during correct (*green*) and incorrect (*red*) trials. Greater proportions of principal cells exhibit larger magnitudes of coherence during correct trials compared to incorrect trials. (**b**) The magnitude of coherence during correct trials plotted against the magnitude of coherence during incorrect trials for all principal cells that were coherent to each rhythm during either correct or incorrect trials. (**c**) The average firing rate during correct trials plotted against the average firing rate during incorrect trials for all principal cells that were coherent to each rhythm during either correct or incorrect trials.

The following figure supplement is available for figure 4:

**Figure supplement 1.** The phase of principal cell coherence to each rhythm during correct and incorrect trials.

rank test Z = 9.85, p<0.00001; Mdn$_{low\ gamma-correct}$ = 0.0231, Mdn$_{low\ gamma-incorrect}$ = 0.0137, Wilcoxon signed-rank test Z = 7.67, p<0.00001; Mdn$_{high\ gamma-correct}$ = 0.0190, Mdn$_{high\ gamma-incorrect}$ = 0.0126, Wilcoxon signed-rank test Z = 7.16, p<0.00001). To ensure that these changes in coherence strength were not due to decreases in principal cell firing rates during incorrect trials, we compared the firing rates across trial types and found no significant differences between correct and incorrect trials (*Figure 4c*; Mdn$_{correct}$ = 0.6000 Hz, Mdn$_{incorrect}$ = 0.4444 Hz, Wilcoxon signed-rank test = 0.5608, p=0.5750). These results further demonstrate that the manner in which principal cells are engaged in each of the four rhythms is strongly related to task performance.

To examine whether performance dependent engagement of the principal cells coincides with a rhythmic phase preference, we compared the average phase of spiking during correct and incorrect trial types for each neuron (*Figure 4—figure supplement 1*, see Materials and methods). The principal cells coherent to theta$_{4-12\ Hz}$ and high gamma$_{65-90\ Hz}$ during Correct Trials Only did not exhibit a consistent phase preference across correct and incorrect trial types (*Figure 4—figure supplement 1c*; $R_{theta-correct}$ = -0.05, p=0.4107; $R_{high\ gamma-correct}$ = -0.10, p=0.2239), suggesting that these principal cells are engaged by theta$_{4-12\ Hz}$ and high gamma$_{65-90\ Hz}$ rhythmic circuits differently on correct and incorrect trials. For the principal cells coherent to beta$_{15-35\ Hz}$ during Correct Trials Only, the preferred phase during correct trials was anti-correlated to the average phase during incorrect trials ($R_{beta-correct}$ = -0.35, p=0.0027), indicating that the preferred phase of entrainment during correct trials in this population was often opposite (i.e. separated by 180 degrees) to the average phase during incorrect trials. In contrast, the principal cells coherent to low gamma$_{35-55\ Hz}$ during Correct Trials Only exhibited similar phases across correct and incorrect trials ($R_{low\ gamma-correct}$ = 0.35, p=0.0006), indicating that there is some similarity in how the spike timing of these cells relates to the low gamma$_{35-55\ Hz}$ oscillation during both trial types despite a lack of significant coherence during incorrect trials. This suggests that the low gamma$_{35-55\ Hz}$ rhythmic circuit tends to engage its principal cells at a specific phase irrespective of trial outcome, while the magnitude of this engagement is a stronger predictor of correct performance. For the relatively few principal cells that were coherent to theta$_{4-12\ Hz}$, beta$_{15-35\ Hz}$, low gamma$_{35-55\ Hz}$, and high gamma$_{65-90\ Hz}$ during All Trials, the preferred phase angles during correct trials were not correlated to the preferred phase angles during incorrect trials (*Figure 4—figure supplement 1d*; $R_{theta-all}$ = -0.12, p=0.3693; $R_{beta-all}$ = -0.89, p=0.1747; $R_{low\ gamma-all}$ = 0.07, p=0.8175; $R_{high\ gamma-all}$ = -0.05, p=0.8751), indicating that these principal cells are engaged by rhythmic circuits differently on correct and incorrect trials. These results further demonstrate that principal cells are distinctly engaged in rhythmic circuits when the rat is effectively processing information in order to correctly respond in the task.

## Interneurons engage in multiple rhythms during successful task performance

We then examined the extent to which interneurons and principal cells were exclusively coherent to one rhythm or flexibly engaged in many. In our previous analyses, cells were identified as coherent to one rhythm without regard for its coherence to the other three frequency ranges examined. It is possible that many of the cells coherent to beta$_{15-35\ Hz}$, for example, are also coherent to theta$_{4-12\ Hz}$, low gamma$_{35-55\ Hz}$, and high gamma$_{65-90H}$. To examine the extent to which cells categorized as coherent to a specific rhythm are actually engaged in many, we first identified the number of interneurons and principal cells coherent to theta$_{4-12\ Hz}$, beta$_{15-35\ Hz}$, low gamma$_{35-55\ Hz}$, or high gamma$_{65-90\ Hz}$ that were either coherent to only one rhythm or multiple rhythms (*Figure 2e*). As the strength of coherence for both interneurons and principal cells is notably lower during incorrect trials, this analysis was restricted to Correct Trials Only. Tests for differences between the interneuron and principal cell populations revealed that interneurons were significantly more engaged in multiple rhythms than the principal cell population in each rhythmic category ($\chi^2_{theta\ (P\ v\ I)}$ (1, N=300) = 49.65, p<0.00001; $\chi^2_{beta\ (P\ v\ I)}$ (1, N=122) = 33.40, p<0.00001; $\chi^2_{low\ gamma\ (P\ v\ I)}$ (1, N=145) = 26.56, p<0.00001; $\chi^2_{high\ gamma\ (P\ v\ I)}$ (1, N=178) = 31.17, p<0.00001; for the interneurons and principal cells engaged in multiple rhythms, see the specific combination of rhythms in *Figure 2—source data 1 and 2*, respectively). Thus, interneurons often participate in multiple types of rhythmic circuits, whereas principal cells often exhibit engagement limited to a single rhythm. These results suggest that different mechanisms may be involved in engaging interneurons and principal cells in rhythmic circuits, leading to the capacity for interneurons to readily participate in multiple rhythmic processing states.

Since the majority of interneurons are coherent to multiple rhythms during correct trials, it is possible that engagement in specific combinations of rhythms is important for successful performance of the task. To investigate this question, we determined the number of interneurons coherent to every combination of the four rhythmic categories during correct and incorrect trials (*Figure 5a*). Interneurons were not equally distributed across the fifteen combinations of the four rhythms during correct trials ($\chi^2_{correct}$ (14, N=134) = 326.97, p<0.00001), with the largest number of the interneurons coherent to all four rhythms. In contrast, during incorrect trials, the interneurons were unequally distributed such that they were most often coherent to theta$_{4-12\ Hz}$ only ($\chi^2_{incorrect}$ (14, N=100) =

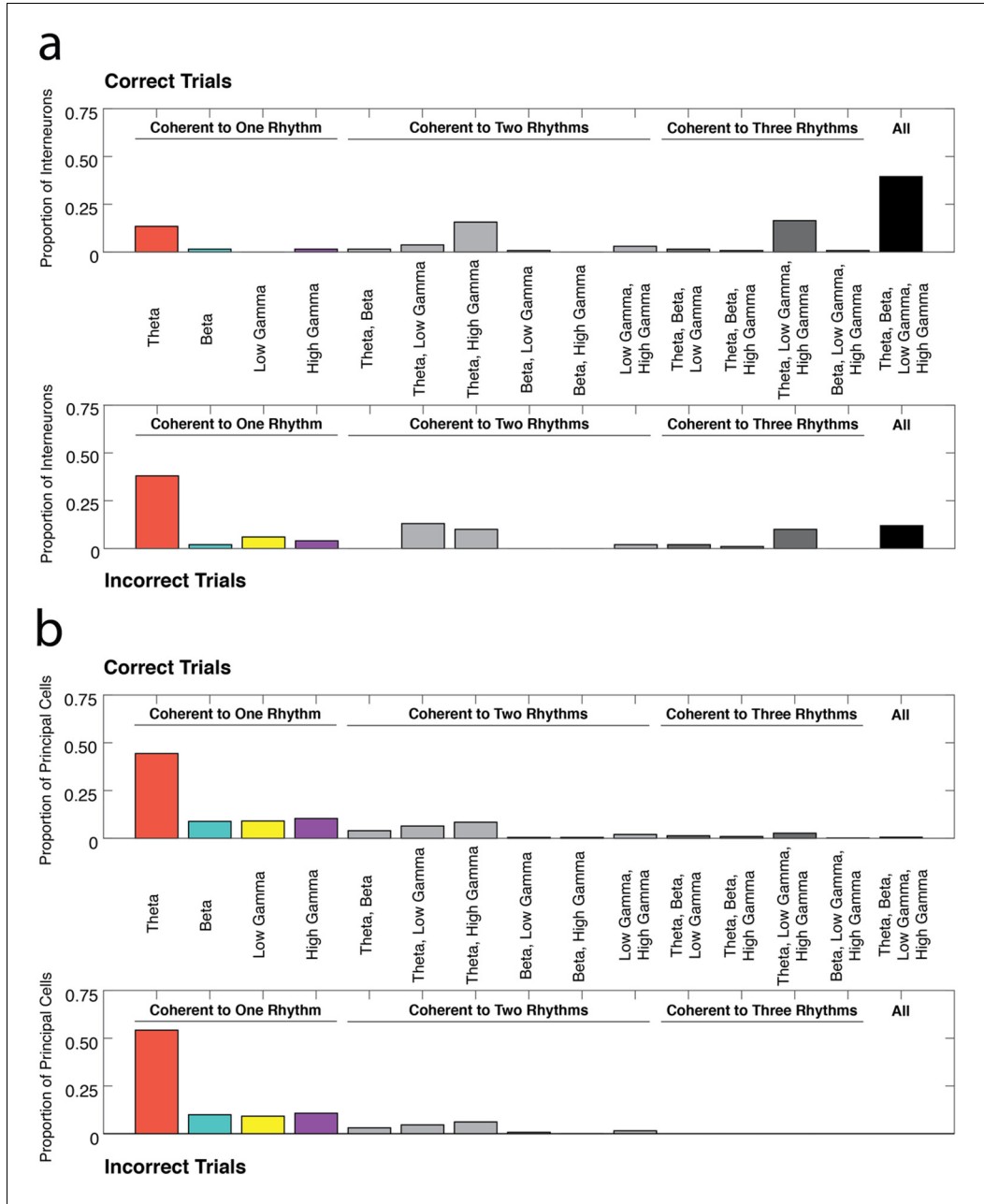

**Figure 5.** Profiles of interneuron and pyramidal cell recruitment into rhythmic circuits during correct and incorrect performance. (**a**) The proportions of interneurons coherent during correct trials (*top*) or incorrect trials (*bottom*) that were coherent to each combination of the four rhythms examined in this study (theta$_{4-12\ Hz}$, beta$_{15-35\ Hz}$, low gamma$_{35-55\ Hz}$, and high gamma$_{65-90\ Hz}$). While a large proportion of interneurons demonstrate coherence to all four frequency ranges during correct trials, interneurons are often engaged in a single rhythmic circuit during incorrect trials. (**b**) Same as in **a**, for principal cells. The majority of principal cells exhibited coherence to a single rhythmic circuit during both correct and incorrect trial types. To view the number of neurons coherent during correct trials as a ratio of all the neurons coherent in each rhythmic category see *Figure 5—figure supplement 1*.

The following figure supplements are available for figure 5:

**Figure supplement 1.** The ratio of neurons coherent during correct trials to all coherent neurons in each rhythmic category.

**Figure supplement 2.** Correlations of interneuron coherence for different pairs of rhythms.

**Figure supplement 3.** Amplitude correlations for different pairs of rhythms.

203.30, p<0.00001). The distribution of interneurons across the fifteen rhythmic categories was significantly different during correct and incorrect trial types ($\chi^2_{correct\ v\ incorrect}$ (14, N=234) = 52.46, p<0.00001). The most striking differences were observed in the proportion of interneurons coherent to theta only and the proportion of interneurons coherent to all four rhythms ($\chi^2_{theta\ only}$ (1, N=234) = 18.99, p<0.00001; $\chi^2_{all\ rhythms}$ (1, N=234) = 21.67, p<0.00001). These results indicate that interneuron engagement in all four rhythms is strongly related to successful performance of the task, and suggests that interneuron engagement solely in a theta rhythmic circuit is a marker of a processing state that is maladaptive for good performance in our task.

Principal cells were also not equally distributed across the fifteen combinations of the four rhythms during correct trials (*Figure 5b*, $\chi^2_{correct}$ (14, N=453) = 1162.70, p<0.00001), with the largest numbers of principal cells coherent to single rhythms and more specifically to theta$_{4-12\ Hz}$ only. A similar bias in the distribution was observed during incorrect trials ($\chi^2_{correct}$ (14, N=131) = 518.35, p<0.00001), with no significant differences across the fifteen rhythmic categories between correct and incorrect trial types. Thus, whereas larger numbers of principal cells are engaged in rhythmic activity during correct trials (*Figure 2b*), co-participation in multiple rhythms is not related to task performance ($\chi^2_{correct\ v\ incorrect}$ (14, N=584) = 12.13, p=0.5958).

## Rhythmic co-modulation during correct trials

While the interneurons engaged in all four rhythms over the course of the session may have been independently engaged by each rhythmic circuit, it is possible that some rhythmic circuits operate cooperatively. If the latter possibility were the case, we would expect the coherence of the interneurons to be similarly modulated on each trial by our rhythms of interest. To examine whether interneurons were engaged in combinations of rhythms at similar times during the nose poke, we performed multi-taper spike-phase coherence analysis to acquire the magnitude of coherence at four frequencies (7 Hz, 20 Hz, 45 Hz, and 75 Hz) within each of the four bands used in this study for every correct trial. We then asked whether the magnitude of coherence to one frequency was correlated with the magnitude of coherence to another frequency across trials by performing correlations on the coherence values across trials for every pair of frequencies (*Figure 5—figure supplement 2*). There were no significant differences in the correlations values obtained for any pair of rhythms for the subpopulation of interneurons that were coherent to all four rhythms (one-way repeated measures ANOVA, N=53, d.f. = 5, F= 1.59, p=0.1648). Although we are open to the possibility that there is coordinated (e.g. correlated or anti-correlated) coherence to multiple rhythms, there do not appear to be any obvious trends in our data.

We also attempted to examine whether the amplitudes of the four rhythms were co-modulated during correct trial nose pokes. To do this, we examined amplitude dynamics in the local field potential alone, independently of the spiking activity. For every session, we calculated the instantaneous amplitude of each of the four frequency ranges during correct trials and performed correlations on the amplitude values for every pair of frequencies (*Figure 5—figure supplement 3a*, see Materials and methods). We observed significant differences in the correlations between different pairs of frequencies (one-way ANOVA, d.f. = 5, F= 42.06, p<0.00001). Post hoc pairwise comparisons revealed that some pairs of frequencies appear to be more correlated or anti-correlated than others (for the results from all pairwise comparisons, see *Figure 5—figure supplement 3a*). Specifically, theta$_{4-12\ Hz}$ and low gamma$_{35-55\ Hz}$ amplitudes are more anti-correlated than all other pairs of frequencies (Tukey's Honest Significant Difference test, p<0.00001, for all post hoc pairwise comparisons of theta$_{4-12\ Hz}$ and low gamma$_{35-55\ Hz}$ amplitude correlations against all other pairs of frequencies). In contrast, theta$_{4-12\ Hz}$ and beta$_{15-35\ Hz}$ amplitudes, as well as beta$_{15-35H}$ and low gamma$_{35-55\ Hz}$ amplitudes were more correlated than most other pairs of frequencies (Tukey's Honest Significant Difference test, $p_{theta-beta\ v\ theta-low\ gamma}$ < 0.00001, $p_{theta-beta\ v\ theta-high\ gamma}$ < 0.00001, $p_{theta-beta\ v\ beta-low\ gamma}$ = 0.248, $p_{theta-beta\ v\ beta-high\ gamma}$ = 0.020, $p_{beta-low\ gamma\ v\ beta-high\ gamma}$ = 0.676; $p_{beta-low\ gamma\ v\ theta-low\ gamma}$ < 0.00001, $p_{beta-low\ gamma\ v\ theta-high\ gamma}$ < 0.00001, $p_{beta-low\ gamma\ v\ beta-high\ gamma}$ < 0.00001, $p_{beta-low\ gamma\ v\ low\ gamma-high\ gamma}$ = 0.003). Correlations between high gamma$_{65-90\ Hz}$ amplitude and other frequencies (theta$_{4-12\ Hz}$, beta$_{15-35\ Hz}$, and low gamma$_{35-55\ Hz}$) were lower than beta$_{15-35\ Hz}$ and low gamma$_{35-55\ Hz}$ correlations, indicating that co-modulation of frequencies with high gamma$_{65-90\ Hz}$ is generally not as strong. These findings suggest that while theta$_{4-12\ Hz}$ and low gamma$_{35-55\ Hz}$ amplitudes tend to undergo opposing changes, they both are often co-modulated with beta$_{15-35H}$ amplitude at some point during correct nose pokes. Importantly, many pairs of

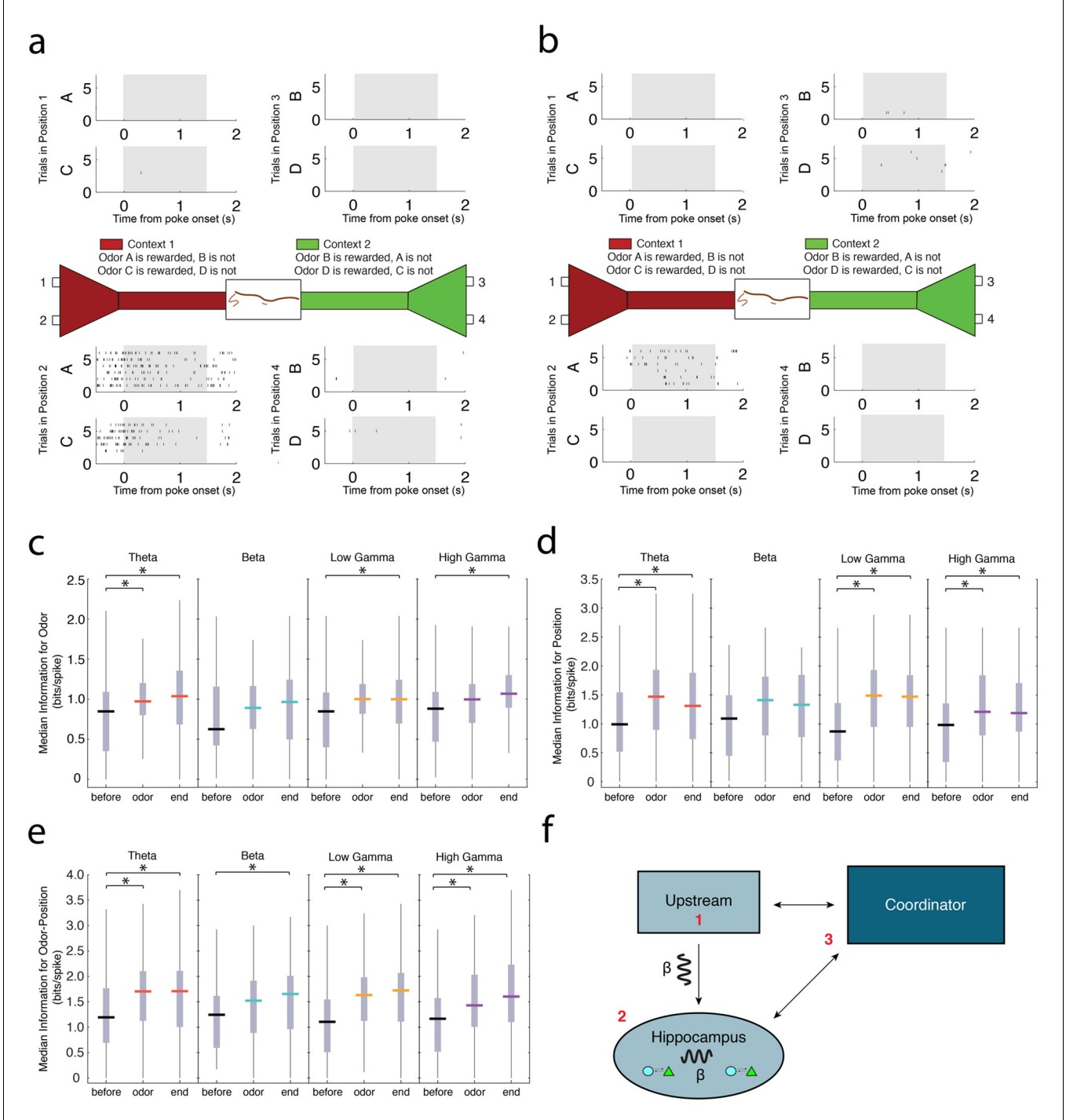

**Figure 6.** Principal cells exhibiting strong spike-phase coherence during the odor sampling contain information for task-relevant features. (a) Representative activity of a single CA1 principal cell during the odor sampling intervals of Correct Trials Only. Each row of tick marks represents spiking during a single trial. This cell demonstrates activity selective for position 2. (b) Same as in a, showing the activity of a principal cell that is selective for the co-occurrence of odor A in position 2. For additional examples of theta$_{4-12 Hz}$, beta$_{15-35 Hz}$, low gamma$_{35-55 Hz}$, and high gamma$_{65-90 Hz}$ coherent principal cells, see *Figure 6—figure supplement 1–4*. (c) Median information (bits/spike) in theta$_{4-12 Hz}$, beta$_{15-35 Hz}$, low gamma$_{35-55 Hz}$, and high gamma$_{65-90 Hz}$ coherent populations for odors during a 500 ms interval prior to nose poke (*before*), 500 ms directly after odor delivery (*odor*), and 500 ms prior to the end of the nose poke when the rat committed to a decision (*end*). Vertical gray bars indicate the inter-quartile range. The top vertical line indicates $q_3 + 1.5 \times (q_3 - q_1)$ and the bottom vertical line indicates $q_1 - 1.5 \times (q_3 - q_1)$, where $q_1$ and $q_3$ are the 25th and 75th percentiles, respectively. Asterisks (*) indicate a significant pair-wise comparison using a Tukey's Honest Significant Difference test, p<0.05. (d) Same as in c, for position information. (e) Same as in c, for odor-position information. (f) Cartoon diagram indicating potential mechanisms for the generation of the

*Figure 6. Continued*

beta$_{15-35 Hz}$ rhythm in CA1: 1) beta$_{15-35 Hz}$ rhythmic input is received from an upstream structure, 2) beta$_{15-35 Hz}$ is an internally generated rhythm, or 3) long-range communication across multiple interacting networks is facilitated by coordination in beta$_{15-35 Hz}$ by a third-party structure.

The following figure supplements are available for figure 6:

**Figure supplement 1.** Theta (4–12 Hz) coherent principal cell.

**Figure supplement 2.** Beta (15–35 Hz) coherent principal cell.

**Figure supplement 3.** Low Gamma (35–55 Hz) coherent principal cell.

**Figure supplement 4.** High Gamma (65–90 Hz) coherent principal cell.

frequencies appear correlated in some sessions and anti-correlated in others (*Figure 5—figure supplement 3b*). Thus, while these results reveal that there are trends towards co-modulation of some frequencies, we did not observe strikingly robust co-modulation or mutual exclusivity for any pair of frequencies in this study. Consequently, it seems that each rhythmic circuit can operate independently, and therefore the engagement of interneurons into each rhythmic network also likely occurs independently.

## Principal cells coherent to each rhythm differentially represent task dimensions

Principal cells in the hippocampus have been shown to exhibit activity that is highly selective to spatial positions, sensory stimuli, and the co-occurrence of these features during associative memory tasks (*Figure 6a,b*, *Figure 6—figure supplement 1–4*) (*Komorowski et al., 2009*; *O'Keefe and Dostrovsky, 1971*; *Eichenbaum et al., 1999*). Consequently, we characterized the degree to which rhythmic populations of principal cells encode specific task dimensions. We calculated the information for odors, positions, or odor-position conjunctions contained in the spiking activity of theta$_{4-12 Hz}$, beta$_{15-35 Hz}$, low gamma$_{35-55 Hz}$, or high gamma$_{65-90 Hz}$ coherent principal cells during three non-overlapping time intervals during correct trials: a baseline 500 ms interval prior to the nose poke (*before*), a 500 ms directly after odor delivery (*odor*), and 500 ms at the end of the nose poke when the rat committed to a decision (*end*; *Figure 6c–e*, for information analysis see Materials and methods) (*Markus, et al., 1994*). Position information describes the extent to which the spiking activity of a neuron differentiates between the four possible odor port positions. Odor information describes the ability of the neuron to discriminate between odors (A, B, C, and D). Odor-position information describes the extent to which neural activity was selective for the co-occurrence of a specific odor in a particular odor port location. A neuron was considered to have significant information for a task dimension if the information score exceeded the 95% confidence interval of 1000 scores calculated from trial-shuffled conditions.

We observed principal cells coherent to each of the four rhythms that also exhibited significant information for task dimensions (*Figure 6—figure supplement 1–4c*). We then tested whether the information for a specific task dimension changed over the course of a trial. For principal cells coherent to each rhythm that also exhibited significant information for a specific task dimension, we compared the information content during the *before, odor,* and *end* intervals described above. Theta$_{4-12 Hz}$ coherent cells exhibited increases in information for each task dimension across the three intervals examined (*Figure 6c–e*, *position*: Friedman's test$_{theta}$: d.f. = 2, $\chi^2$ = 47.19, p<0.00001; *odor*: Friedman's test$_{theta}$: d.f. = 2, $\chi^2$ = 15.89, p=0.0004; *odor-position*: Friedman's test$_{theta}$: d.f. = 2, $\chi^2$ = 31.54, p<0.00001). Post hoc comparisons revealed that these increases occurred after the odor delivery, and lasted until the end of the nose poke (Tukey's Honest Significant Difference test, p<0.05 for comparisons of the *before* interval to *odor* and *end* intervals). Theta$_{4-12 Hz}$ coherent cells thus contained greater information during intervals after odor onset than during approach. Low gamma$_{35-55 Hz}$ and high gamma$_{65-90 Hz}$ coherent cells also exhibited increases in information across the three intervals examined (low gamma: *position*: Friedman's test$_{low\ gamma}$: d.f. = 2, $\chi^2$ = 33.33, p<0.00001; *odor*: Friedman's test$_{low\ gamma}$: d.f. = 2, $\chi^2$ = 5.93, p=0.05; *odor-position*: Friedman's

test$_{\text{low gamma}}$: d.f. = 2, $\chi^2$ = 33.73, p<0.00001; high gamma: *position*: Friedman's test$_{\text{high gamma}}$: d.f. = 2, $\chi^2$ = 20.56, p<0.00001; *odor*: Friedman's test$_{\text{high gamma}}$: d.f. = 2, $\chi^2$ = 7.93, p=0.0189; *odor-position*: Friedman's test$_{\text{high gamma}}$: d.f. = 2, $\chi^2$ = 26.7, p<0.00001). Post hoc comparisons revealed that the increases for position and odor-position information were sustained during both intervals after the odor onset, while information for odors increased only at the end of the poke (Tukey's Honest Significant Difference test, p<0.05). In contrast, beta$_{15\text{-}35 \text{ Hz}}$ coherent cells exhibited significant increases in information only for odor-position conjunctions, and only at the end of the poke, when the rat had committed to a decision (*position*: Friedman's test$_{\text{beta}}$: d.f. = 2, $\chi^2$ = 5.78, p=0.0556; *odor*: Friedman's test$_{\text{beta}}$: d.f. = 2, $\chi^2$ = 2.96, p=0.2275; *odor-position*: Friedman's test$_{\text{beta}}$: d.f. = 2, $\chi^2$ = 6.56, p=0.0376). Together, these results suggest that rhythmic circuits differentially contribute task-relevant information, and that these contributions occur during different key intervals of the task. Moreover, beta$_{15\text{-}35 \text{ Hz}}$ rhythmic circuits may be particularly selective for processing odor-position information, which is critical for successful task performance.

## Discussion

Neural oscillations provide insight into organized interactions between cells at both local circuit and cross-regional scales. Understanding the flexible engagement of neurons into rhythmic circuits allows us to uncover the mechanisms through which single cells contribute to systems level processes. Our results indicate that the hippocampus can support multiple distinct, rhythmically identifiable processing states that are tightly linked to behavior in a context-guided odor-reward association task. During odor-sampling intervals, we observed transient amplitude dynamics in theta (4–12 Hz), beta (15–35 Hz), low gamma (35–55 Hz), and high gamma (65–90 Hz) oscillations, indicating that there is a shift in processing state within the hippocampal circuit. We suggest that this shift serves to coordinate local neural activity for the processing of task-relevant information.

We found that task-related processing coincided with the engagement of interneurons into local rhythmic circuits. We quantified this engagement by determining which interneurons demonstrated significant spike-phase coherence to the ongoing rhythms during odor sampling, as well as the magnitude and phase of their coherence. These measures demonstrated that engagement arises from a reorganization of spike timing with respect to each rhythm during task-related processing (*Figure 2a*, *Figure 3a and b*), and not from enhanced participation through differences in overall firing rate between correct and incorrect trials (*Figure 3c*). Importantly, the relationship between task performance and interneuron engagement differed across the four frequency ranges examined, suggesting that coordination within each rhythm may have a differential contribution to task-related processing. Notably, the largest proportion of interneurons exhibiting significant spike-phase coherence to theta$_{4\text{-}12 \text{ Hz}}$ did so irrespective of whether the trial was correct or incorrect (*Figure 2a*), suggesting that recruitment of these interneurons into theta$_{4\text{-}12 \text{ Hz}}$ rhythmic circuits is not related to successful performance of the task. In contrast, interneurons that exhibited significant spike-phase coherence to beta$_{15\text{-}35 \text{ Hz}}$ were coherent almost exclusively during correct trials, suggesting that task-related processing consistently recruits interneurons into beta$_{15\text{-}35 \text{ Hz}}$ rhythmic circuits. Thus coordination of interneuron activity within each rhythm corresponds to task-related processing to a different extent. We further characterized task-related coordination in our analysis of phase preferences during correct and incorrect trials (*Figure 3—figure supplement 1*). We found that the phase preference between trial types was grossly different for the class of interneurons coherent during Correct Trials Only, but not for the interneurons that were coherent during All Trials. This indicates that task-related engagement manifests in both magnitude and preferred phase, suggesting that interneuron spike timing relative to the ongoing rhythms reflects its participation in task-related processing states. As the proportions of cells coherent during Correct Trials Only and All Trials differ across the four rhythms, we can surmise that each rhythm contributes uniquely to the orchestration of spike timing in the service of task-related processing.

The task-related engagement of the principal cells into rhythmic circuits was strikingly different than the engagement of interneurons. Notably, principal cell engagement in each rhythm occurred almost exclusively during correct trials (*Figure 2b*, *Figure 4a and b*), despite similar firing rates between correct and incorrect trials (*Figure 4c*). This could be observed through the large proportions of principal cells with significant spike-phase coherence to each rhythm during Correct Trials Only (*Figure 3a*). Selective engagement of the principal cell population during correct trials was not

equal across the four frequencies examined. Beta$_{15-35\ Hz}$, low gamma$_{35-55\ Hz}$, and high gamma$_{65-90\ Hz}$ coherent principal cells exhibited this preferential coherence during Correct Trials Only more often than theta$_{4-12\ Hz}$ coherent principal cells, which consisted of a larger proportion of cells that were coherent irrespective of trial outcome. Though this difference is less pronounced in theta rhythmic principal cells than theta rhythmic interneurons, it nonetheless suggests that task-related processing more consistently recruits principal cell activity in beta$_{15-35\ Hz}$, low gamma$_{35-55\ Hz}$, and high gamma$_{65-90\ Hz}$ rhythmic circuits than in theta$_{4-12\ Hz}$ rhythmic circuits. The strength of principal cell coherence to each rhythm was also greater during correct trials than during incorrect trials (*Figure 4a and b*), suggesting that there is an organization of principal cell spike timing that is unique to correct trials. The analysis of phase preference between correct and incorrect trials also revealed unique spike timing properties across the rhythms. The principal cells coherent to theta$_{4-12\ Hz}$, beta$_{15-35\ Hz}$, and high gamma$_{65-90\ Hz}$ during Correct Trials Only exhibited inconsistent phases of entrainment across correct and incorrect trials, providing further evidence that these principal cells are engaged in their rhythmic circuits differently on correct and incorrect trials. In contrast, the principal cells coherent to low gamma$_{35-55\ Hz}$ during Correct Trials Only exhibited similar phases across correct and incorrect trials, suggesting that there are features of engagement in low gamma$_{35-55\ Hz}$ rhythmic circuits that are independent of trial outcome. Despite these finer differences in the nature of principal cell coordination within each rhythmic circuit, principal cells demonstrate a selective reorganization of spike timing during task-related processing.

The extent of engagement in multiple rhythms also differed between the two cell types. Interneurons flexibly interacted in multiple rhythmic circuits, which was most apparent in the large proportion of interneurons that exhibited coherence to all four rhythms during correct trials, though potentially not all at the same time (*Figure 5a*). Indeed, interneuron spike-phase coherence to multiple rhythms might be a signature of correct performance, whereas interneuron engagement in a single rhythm is more often observed during incorrect trials. This result indicates that the flexible engagement of the interneuron population in multiple rhythmic circuits is important for task-related processing. In contrast, larger numbers of principal cells were preferentially coherent to only one rhythm (*Figure 5b*), providing evidence that theta$_{4-12\ Hz}$, beta$_{15-35\ Hz}$, low gamma$_{35-55\ Hz}$ and high gamma$_{65-90\ Hz}$ reflect functionally distinct processing states. This result suggests that principal cells can often participate in segregated rhythmic circuits with separable functions. Moreover, as many principal cells were engaged in single rhythmic circuits during correct and incorrect trials, engagement in multiple rhythmic circuits appears more related to task performance in the interneuron population. These dramatically different profiles of engagement between interneurons and principal cells indicate that each group interacts with the surrounding rhythmic circuits in markedly different ways.

The engagement of interneurons in multiple rhythmic circuits could indicate co-modulation of interneuron activity in more than one rhythmic circuit at the same time. Although the strength of interneuron coherence did not appear to be co-modulated for any pair of rhythms (*Figure 5—figure supplement 2*), correlations in amplitude between pairs of rhythms can indicate potential for co-modulation in the network. The instantaneous amplitudes of certain pairs of rhythms were correlated during the task, although not consistently across all sessions. Specifically, beta$_{15-35\ Hz}$ amplitude was often correlated with theta$_{4-12\ Hz}$ and low gamma$_{35-55\ Hz}$ amplitude, revealing a potential for beta$_{15-35\ Hz}$ coherent interneurons to be co-modulated by theta$_{4-12\ Hz}$ and low gamma$_{35-55\ Hz}$ rhythmic circuits. Interestingly, theta$_{15-35\ Hz}$ amplitude was often anti-correlated with low gamma$_{35-55\ Hz}$ amplitude, suggesting that interneuron engagement in theta$_{4-12\ Hz}$ and low gamma$_{35-55\ Hz}$ rhythmic circuits may occur at different time points during the nose poke interval. These coordinated dynamics among specific pairs of rhythms suggest that there is some consistency in the organization of interneuron activity within each of the four rhythms that can include both co-modulation within some rhythmic circuits and temporally segregated entrainment in others.

The engagement of principal cells in single rhythmic circuits may provide a mechanism for the hippocampus to simultaneously engage subpopulations of cells in distinct task-relevant processes. It is possible that subpopulations of principal cells become differentially engaged in their surrounding rhythmic circuits as a consequence of subtle differences in innervation from heterogeneous afferents or differences in the narrow-band intrinsic resonance of CA1 principal cells (*Freund and Buzsáki, 1996*; *Stark et al., 2013*.). Task-related increases in coherence could then facilitate the effective communication of subpopulations of CA1 neurons with specific downstream targets that exhibit similar resonance. The ability of a single network to process information at multiple different frequencies

thus creates an opportunity for multiplexing through frequency division and the selective readout of signals that are segregated into different frequency bands (*Akam and Kullmann, 2014.*). This feature could be particularly useful in the hippocampus for combining input from multiple different afferents while maintaining the ability to transmit individual signals.

We demonstrate that the principal cells coherent to each rhythm can contribute task-relevant information at different time intervals during the odor-sampling epoch (*Figure 6c–e*), providing further evidence that each rhythm reflects a distinct circuit process. Theta$_{4-12 Hz}$, low gamma$_{35-55 Hz}$ and high gamma$_{65-90 Hz}$ coherent principal cells exhibited significant increases in information for position and odor-position after odor delivery that lasted until the end of the nose poke. Information for odors increased earlier in theta$_{4-12 Hz}$ coherent principal cells than in low gamma$_{35-55 Hz}$ and high gamma$_{65-90 Hz}$ coherent principal cells, indicating that theta$_{4-12 Hz}$ coherent principal cells have a dissociable contribution to the processing of odor information. Notably, beta$_{15-35 Hz}$ coherent principal cells only exhibited increases in information for odor-position conjunctions, which are a hallmark of associative memory (*Komorowski et al., 2009*). Coupled with the fact that interneurons are coherent to beta$_{15-35 Hz}$ primarily during correct trials (*Figure 2a*), our results strongly suggest that beta$_{15-35 Hz}$ rhythmic circuits might be uniquely processing information that is critical for successful utilization of associative memory.

Multiple mechanisms could generate beta$_{15-25 Hz}$ rhythmicity within the hippocampal network. Identifying the mechanisms that give rise to the beta$_{15-35 Hz}$ rhythm can provide insight into how it contributes to associative memory processes. The selective entrainment of the interneuron population to beta$_{15-25 Hz}$ during correct trials (*Figure 2a*) could reflect the receipt of rhythmic input from an upstream structure that engages neurons in the hippocampus (*Figure 4d*, mechanism **1**). This hypothesis is supported by recent studies suggesting that coherence between the CA1 and the lateral entorhinal cortex (LEC) parallels the onset of learning in an olfactory discrimination task and the development of task-selective ensemble activity (*Igarashi et al., 2014*; *Rangel and Eichenbaum, 2014*). As the LEC has been shown to be important for the processing of multi-modal object information (*Young et al., 1997*; *Deshmukh and Knierim, 2011*), communication between LEC and hippocampus could be critical for the association of odors with their reward contingencies. In addition, a recent study that used current source density analysis to reveal the location of current sources and sinks corresponding to beta oscillations in the dentate gyrus found that beta oscillations in this structure are likely driven by perforant path input (*Rangel et al., 2015*). It is thus possible that beta oscillatory dynamics in the CA1 are the product of entrainment by an upstream cortical afferent. As an alternative to inheriting beta$_{15-25 Hz}$ rhythmicity from upstream structures, it is also possible that the hippocampus locally generates a beta$_{15-25 Hz}$ rhythm when engaged by specific afferents (*Figure 4d*, mechanism 2). Under this hypothesis, a change in communication between local neurons produces a functional circuit that resonates at beta$_{15-25 Hz}$, which optimally facilitates associative memory processing. As a third possibility, the beta$_{15-25 Hz}$ rhythm could reflect the broad coordination of activity across disparate neural networks (*Kopell et al., 2000*; *Bibbig et al., 2002*; *Pinto et al., 2003*). The hippocampus is just one of many structures that engage in beta rhythmic processing during the presentation of meaningful cues (*Igarashi et al., 2014*; *Kay and Freeman, 1998*; *Rangel et al., 2015*; *Quinn et al., 2010*; *Buschman et al., 2012*; *Leventhal, 2012*; *Tingley et al., 2015*). Together, these structures could constitute an integrated network of circuits that span the brain. Rhythmic synchronization of this distributed network through a central rhythm generator could then enable information processing as a coordinated unit (*Figure 4d*, mechanism 3).

Our study provides insight into the flexible coordination and engagement of distinct cell types within hippocampal circuits. We found two major differences in the rhythmic organization of CA1 interneuron and principal cell activity during a memory task. First, interneuron and principal cell coherence have distinct relationships to task performance. Second, we show that interneurons, unlike principal cells, are often flexibly engaged in multiple rhythms. These differences may shed light upon the distinct roles of excitation and inhibition in processing within rhythmically identifiable hippocampal circuits. Taken together, our results suggest that different rhythms make unique contributions to information processing within the hippocampus, and changes in the rhythmic profile reflect dynamic coordination of its cell activity. Further characterization of these rhythmic circuits will be critical for understanding how cell activity within the hippocampus is flexibly coordinated in the service of memory.

# Materials and methods

## Rats

All animal procedures were performed in accordance with NIH and Boston University Institutional Animal Care and Use Committee guidelines. Subjects were six male Long-Evans rats (Charles River Laboratories) housed individually and maintained on a 12-hr light/dark cycle. All neural recordings were performed during the light cycle. Rats were food and water restricted, and maintained at a weight of at least 85–90% of *ad libitum* body weight. Weights ranged from 450-–600g.

## Behavioral training

All rats were handled daily for at least two weeks before beginning the experiment. Upon entering the study, rats were first exposed to the testing apparatus: a two-arm apparatus constructed of black plastic (*Figure 1—figure supplement 1*). Two 45-cm arms extended from opposite ends of a 30-cm central chamber. The central chamber consisted of 20-cm high walls and two pairs of doors, one black and one clear, that opened onto either arm. All doors could be independently raised and lowered by electric actuators. The end of each arm contained a widened area with two circular odor ports (*Figure 1—figure supplement 1*, inset), into which an odorant could be released by opening an air solenoid. Odorants were delivered by air flowing over vials of oil-based scents. A total 12 odors were used over the course of training and experimental sessions. Some of the odors were natural scents (maple, cedar, spearmint, strawberry, sweet orange, mango, lemon) while the others were chemical odorants (2-phenylpropionaldehyde, allyl-a-ionone, cis-3-hexen-1-ol, guaiacol, isoamyl acetate). Each port also contained a vacuum, which removed the odorant after release to prevent cross-contamination of odors between trials. Below each odor port was a tray with a well, into which water could be released to reward successful performance. Throughout the apparatus there were LED sensors to verify rat movement: two along the length of each arm, one in each odor port, one in each water well, and two in the central chamber. LED sensors in each odor port acted to record nose poke onset. All apparatus functions were controlled via a computer by customized MATLAB programs (MathWorks, Natick, MA).

Each rat underwent a behavioral shaping process to first poke its snout into an odor port and then increase the length of each nose poke. Rats received training sessions in which initially 100 ms nose pokes elicited a water reward. The nose poke criterion increased by 100 ms for each poke of sufficient length and dropped by 100 ms for every two successive unsuccessful nose pokes until rats learned to consistently poke for 1.5 s. During this process, rats only had access to one port at a time and received equal exposure to all four ports on the apparatus. Following nose poke training, each rat was taught to discriminate between two odors of an odor pair. During early discrimination sessions, two different context overlays made of distinct materials were placed over each side of the apparatus, and the rat was given access to one arm of the apparatus at a time. Rats alternated between arms in blocks of 20 trials during initial training, followed by blocks of 10 trials upon improved performance. Ultimately, arms switched in a pseudorandom, counterbalanced fashion in all later sessions.

## Full task

During each trial, each of the odors would be presented on one side of the apparatus, one odor in each of the two ports. Rats could initiate the delivery of an odor by poking their snouts into an odor port. Odors were released within the port after a 250 ms delay. One odor of the pair was designated as a "correct" odor, and sustaining a nose poke of 1.5 s in the odor port containing this odor resulted in a water reward. The other odor was not rewarded, and a white noise buzz would occur if the rat sustained a nose poke for 1.5 s in that port. After each trial, the rat returned to the central chamber, the doors surrounding the chamber were elevated, and the next trial would begin. We analyzed neural activity during trials when the rat maintained a nose poke for 1.5 s while sampling a rewarded odor (correct trials) and during trials when the rat maintained a nose poke for 1.5 s while sampling the non-rewarded odor (incorrect trials). The same odor was always correct for a given context overlay and side of the apparatus. The location of the correct odor could switch between the left and right ports each trial, and trials were counterbalanced and pseudo-randomized before each session. The reward contingencies of the odors were reversed for each arm: the incorrect odor from

the first arm was the correct odor on the second arm and vice versa. Each rat underwent 80 trials a day until reaching a criterion of 75% accuracy.

With this final paradigm, rats were trained to perform this task with four pairs of odors (eight odors total) in a 96-trial session. Each odor pair was presented in a discrete block of 24 trials. For each recording session, distinct context overlays were placed on the apparatus for every two consecutive odor pairs. That rat was removed from the apparatus when the context overlays were replaced. Each half of the session was analyzed separately. All conditions were counterbalanced and pseudo-randomized within each 24-trial block before each session. For data analysis, only sessions in which the rat performed at 75% accuracy were used.

## Hyperdrive implantation surgery

Following training, each rat was surgically implanted with a hyperdrive containing 24 microdrives, each with an independently drivable tetrode. Each tetrode was composed of four strands of 0.0005″ (12 μm) Nickel-Chrome wire (Sandvik, Stockholm, Sweden), gold-plated to reduce impedance to 200–250 kOhms at 1000 Hz. The implant site was located over the right dorsal hippocampus (A/P = -4.0 mm; M/L = 2.2 mm), and tetrodes were turned down an initial 1.6 mm into the brain immediately following surgery. After rats received a two-week recovery period, tetrodes were progressively lowered over the training period (6–7 weeks) to the principal cell layer of CA1 (D/V = ~1.9 mm).

## Neural recordings

Signals were amplified by a preamplifier 20x and amplified again to 4,000–6,000x (Plexon, Dallas, TX), with a band-pass filter of 400–8,000 Hz to digitally isolate spikes (OmniPlex, Plexon). Local field potentials (LFPs) were digitally isolated with a band-pass filter from 1–400 Hz. LFP and spike channels were globally referenced to a wire above the cerebellum, and spike channels were also locally referenced to a wire with low activity. Throughout the session, the rat's location was recorded via digital video and tracking software (CinePlex, Plexon, Dallas, TX) that monitored the motion of two LEDs mounted at the top of the rat hyperdrive. Tracking data was time stamped and synchronized with neural recording data, all of which was stored offline for later analysis. Only sessions in which the rats performed at greater than 75% accuracy were used in analysis. Electrophysiological features such as the presence of theta (4–12Hz) oscillations, sharp-wave ripples, and theta-modulated complex spiking activity were used to estimate tetrode locations. To confirm tetrode locations, rats were anesthetized with 2.5% isofluorane and small lesions were made by passing 40 μA of direct current through each wire. Final tetrode locations were visualized via a Nissl stain in 40 μm coronal sections. Single units were isolated in OfflineSorter (Plexon) by comparing waveform features across tetrode wires including peak and valley voltage amplitudes, total peak-to-valley distance, and principal component analysis.

Principal cells and interneurons were classified according to both firing rate and waveform characteristics. Interneurons clustered according to mean firing rate, mean width at half the maximum amplitude of the waveform, and mean temporal offset from peak to trough. Interneurons exhibited mean firing rates of at least 5 Hz, a mean width at half-max less than 150 μs, and a mean peak to trough waveform width less than 350 μs, while principal cells exhibited mean firing rates of less than 3 Hz with wider waveforms (*Stark et al., 2013*; *Csicsvari et al., 2003*; *Bartho, 2004*). Approximately 5–10 recording sessions were performed over the course of 3–5 weeks from each rat, with rats resting on days between sessions.

## Position, odor, and odor-position information analyses

For populations of principal cells coherent to theta$_{4-12 Hz}$, beta$_{15-25 Hz}$, low gamma$_{35-55 Hz}$, and high gamma$_{65-90 Hz}$ during correct trials, we calculated the information for positions, odors, and odor-position conjunctions expressed in their spiking activity. All information scores were calculated using the spiking activity from correct trials only. Position information describes the extent to which the spiking activity of a neuron differentiates between the four possible odor port positions. Odor information describes the ability of the neuron to discriminate between four odors (A, B, C, or D). Since odors are only rewarded on one side of the maze and we only considered correct trials, odor identity is nested within the identity of a given side of the maze. Odor-position information describes the

extent to which neural activity was selective for the co-occurrence of a specific odor in a particular odor port location. The position information score (*Markus et al., 1994*) was calculated as follows:

$$I = \sum P_i \left(\frac{F_i}{F}\right) \log_2 \left(\frac{F_i}{F}\right)$$

Where $i$ is the odor port position number (four possible positions), $P_i$ is the probability of occupancy in position $i$, $F_i$ is the mean firing rate for position $i$, and $F$ is the overall mean firing rate of the cell.

For the calculation of odor information, a similar formula was used where $P_i$ is the probability of experiencing an odor, and $i$ is the number of the odor (four possible odors). For the calculation of odor position information, $P_i$ is the probability of experiencing an odor in a given position, and $i$ is the number of the odor-position combination (eight possible odor-position combinations for correct responses in two consecutive blocks of odor pairs). To hold reward value constant, odor A was compared at each position with odor C, and B with D.

In order to determine whether calculated scores could be acquired by chance from the spiking behavior of a given principal cell, task conditions were randomly shuffled 1000 times and the observed information was considered significant if greater than the 95% confidence interval of the condition-shuffled scores. Principal cells could exhibit information for more than one task dimension. Information scores were then calculated for three different 500 ms intervals: 750 ms-–250 ms prior to nose poke onset (*before*), the 500 ms after odor onset (*odor*), and the last 500 ms of the nose poke (*end*). This latter analysis was performed on all cells that exhibited significant information in their spiking behavior for a given task dimension during the entire duration of correct nose pokes. Differences in the median information across the three time intervals were assessed using a Friedman's test, with post-hoc pairwise comparisons performed using a Tukey's Honest Significant Difference test.

## Spectrograms

Average spectrograms for the odor-sampling intervals of correct and incorrect trials were calculated using a multi-taper method from the Chronux open source MATLAB toolbox (available at: http://chronux.org/) (*Mitra and Bokil, 2008*). Spectrograms were calculated for intervals beginning 0.5 s prior to the nose poke and lasting until 1.5 s after its initiation. The results from multiple correct trials in a single session were averaged and then divided by the mean amplitude observed during 2.0 s baseline inter-trial intervals in the center chamber of the behavioral apparatus. The log of these values was then taken before averaging across all sessions from all rats. For comparison between correct and incorrect trials, the results from multiple correct trials in a single session were averaged and then divided by the amplitude during incorrect trials. The log of these values was then taken before averaging across all sessions from all rats.

## Local field potential amplitude analyses

A 3$^{rd}$-order Butterworth filter was first used to bandpass filter the LFP for theta (4–12 Hz), beta (15–35 Hz), or low gamma (35-–55 Hz), or high gamma (65–90 Hz) (*Rubino et al., 2006*). These frequency ranges were chosen based upon the observed frequencies present in the average spectrogram during odor sampling (*Figure 1*). The instantaneous amplitude for the entire session was then calculated by taking the magnitude of the complex Hilbert transform of the filtered signal. The mean amplitudes during correct and incorrect trials were then calculated by averaging the amplitudes observed during the 1.5 s nose poke intervals leading to a correct or incorrect response for a single session. To examine increases or decreases in amplitude over the course of a nose poke, the mean amplitudes for each session were averaged within six 250 ms time bins spanning the 1.5 s nose poke interval. A two-factor repeated measures ANOVA was used to determine whether amplitudes changed over the course of the six time bins or differed according to behavioral outcome (correct or incorrect).

To examine whether amplitude was co-modulated across rhythms during correct trials of a session, we calculated the instantaneous amplitude of each of the four frequency ranges during correct trials and performed correlations on the amplitude values for every pair of frequencies. For each session, a random temporal jitter was applied to instantaneous amplitude of each rhythm 1000 times to

create a null distribution of correlations that would be expected by chance. All correlations in the data were significantly above the 95% confidence interval of distributions derived in this manner. To determine whether there were any differences in the correlations between pairs of frequencies, we performed a one-way ANOVA with post-hoc pairwise comparisons performed using Tukey's Honest Significant Difference test.

### Spike-Phase coherence analyses

As mentioned above, a 3$^{rd}$-order Butterworth filter was first used to bandpass filter the LFP for theta (4–12 Hz), beta (15–35 Hz), low gamma (35–55 Hz), or high gamma (65–90 Hz) (*Rubino, et al., 2006*). The instantaneous phase was then calculated by taking the arctangent of the complex Hilbert transform of the filtered signal. Single cell spike-phase relationships to the filtered LFP during the odor sampling intervals were assessed using a Rayleigh statistic, and categorized as significantly phase coherent if exhibiting a $p<0.05$. Spike-phase relationships were calculated for correct trials only or incorrect trials only for the 1.5 s prior to a reward or white noise buzz outcome, respectively. The magnitude and phase of coherency for cell spiking activity with respect to each rhythm was calculated using a multi-taper method from the Chronux open source MATLAB toolbox (available at: http://chronux.org/) (*Mitra and Bokil, 2008*), and adjusted for differences in firing rate between correct and incorrect trials by estimating a correction factor that is conceptually equivalent to spike thinning procedures (*Aoi et al., 2015*). The number of tapers ranged from 3–15, to maximize the number of tapers that could be used while avoiding contamination from neighboring frequencies outside the boundaries of a given frequency range. The magnitude and phase of coherence at approximately 7 Hz, 20 Hz, 45 Hz, and 75 Hz were reported for the theta$_{4-12\ Hz}$, beta$_{15-35\ Hz}$, low gamma$_{35-55\ Hz}$, or high gamma$_{65-90\ Hz}$ frequency ranges, respectively. To test whether there was any consistency in the preferred phase angles during correct and incorrect trials, we performed circular correlations on the phases observed across a given population of interneurons or pyramidal cells during each trial type.

For the interneuron population that exhibited significant spike-phase coherence to all four rhythms at some point during correct trials, we tested whether the magnitude of coherence to each rhythm was correlated across trials. For each session, we first determined the magnitude of coherence at 7 Hz, 20 Hz, 45 Hz, and 75 Hz for every correct trial. We then determined whether the magnitude of coherence to one frequency was correlated with the magnitude of coherence to another frequency across trials by performing correlations on the coherence values across trials for every pair of frequencies. To test for significant differences in the correlations between pairs of frequencies, we performed a one-way repeated measures ANOVA.

## Acknowledgements

This work was partially supported by the NSF DMS-1042134 and MH052090. We would like to thank Dr. Benjamin Pittman-Polletta, Dr. Mikio Aoi, Dr. Daniel Gibson, and Dr. Nancy Kopell for helpful conversations. We would also like to thank Jeremiah Rosen, Khushboo Chawla, Brian Ferreri, Catherine Mikkelsen, and Rapeechai Navawongse for technical assistance.

## Additional information

#### Competing interests

HE: Reviewing editor, *eLife*. The other authors declare that no competing interests exist.

#### Funding

| Funder | Grant reference number | Author |
| --- | --- | --- |
| National Science Foundation | DMS-1042134 | Lara M Rangel |
| National Institute of Mental Health | MH052090 | Howard Eichenbaum |

The funders had no role in study design, data collection and interpretation, or the decision to submit the work for publication.

## Author contributions

LMR, JWR, Conception and design, Acquisition of data, Analysis and interpretation of data, Drafting or revising the article; PDR, KRK, BSP, ISH, CHB, Acquisition of data, Analysis and interpretation of data, Drafting or revising the article; HE, Conception and design, Analysis and interpretation of data, Drafting or revising the article

## Author ORCIDs

Blake S Porter, http://orcid.org/0000-0003-1278-0289

## Ethics

Animal experimentation: All animal procedures were performed in accordance with NIH guidelines and approved by the Institutional Animal Care and Use Committee (IACUC) at Boston University, Approval Number: 13-057.

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
