## [Decision Letter]

Thank you for submitting your work entitled "Rhythmic coordination of hippocampal neurons during associative memory processing" for further consideration at *eLife*. Your article has been favorably evaluated by Eve Marder (Senior editor) and three reviewers, one of whom is a member of our Board of Reviewing Editors.

Your study was discussed by the Reviewing editor and two reviewers who came to the conclusion that your manuscript has the potential to be published in *eLife* after a major revision has been performed. Although the reviewers judge your study on the correlation of principal cell and interneuron activity in relation to three neuronal network rhythms in CA1 at beta, low gamma and theta frequencies during a context-guided memory task as important, the reviewers had significant criticisms of the presentation of the results and conclusions. The reviewers ask for major improvements in the way the results are presented in their written form and in form of images. The clarity and explanatory power of the data should be improved, and in some places the reviewers did not feel that the conclusions are adequately supported by the data.

There was some confusion with regard to the statistics. Questions emerged in relation to which groups of data have been compared, whether groups of data may have been pooled and on the chi-square analysis. Finally, reviewer #3 asks important questions concerning the mechanisms by which individual and presumably neighboring cells could become phase locked to different rhythms. Below you will find extracted comments of the three reviewers for your careful consideration in the revision process.

*Reviewer #1:*

The study of Rangel et al. investigates the correlation of principal cell and interneuron activity in relation to three neuronal network rhythms emerging in CA1, beta, low gamma and theta rhythms during a context-guided odor-reward association task.

Generally, the behavioral and electrophysiological part of the study is perfectly performed. The Introduction and Discussion are very well written. The Results section could be improved in the clarity of the presented results. It is a bit difficult to follow the individual parts of the results, in particular, to keep similarities and differences in the neuronal activity during the various brain rhythms separated. The presentation of the results in Figures 2 and 3 could be improved. The differences between PC and IN recruitment among the various brain rhythms is not easy to understand from the figures. The authors could try to speculate a bit more on the mechanisms which may underlie the emergence of beta activity in the Discussion section.

*Reviewer #2:*

1) The chi-square analyses comparing proportions of cells exhibiting significant spike-phase coherence were not very clearly explained in the Results. For example, in the third paragraph, the authors write: "the majority of beta coherent interneurons exhibited their coherence to beta only during correct trials (chi-square beta = 51.54, d.f., = 2, p < 0.00001)". The chi-square presumably compares the correct category with the incorrect and both categories. How were the expected proportions determined? Similar ambiguities exist for other descriptions of chi-square tests. In the same paragraph, the authors write, "substantial proportions of theta and gamma coherent interneurons exhibited coherence during all trials, irrespective of outcome" and then report significant chi-square statistics for theta and gamma. Did these chi-square tests determine whether the proportions of cells that phase-locked to both correct and incorrect trials was higher than expected? If so, this should be clearly explained. Also, the authors write, "the proportions of interneurons that exhibited coherence contingent on trial outcome differed across the three rhythms" and report significant chi-square tests for theta-beta, theta-gamma, and beta-gamma. What proportions were compared across the different rhythms? What exactly does "contingent on trial outcome" mean? Were correct-selective and incorrect-selective proportions grouped together for these tests? I recommend that the authors re-read all of the chi-square results related to Figures 2 and 3 and make sure that readers can easily determine what was compared.

2) The authors examine theta, beta, and low frequency gamma. What about high frequency gamma?

3) In the third paragraph of the Results section, the authors write: "these results indicate that interneuron entrainment to the beta rhythm is related to successful task performance". I was confused by this conclusion since Figure 2 (middle) shows that more interneurons were entrained to only beta during incorrect trials. For correct trials, Figure 2 (middle) shows that most neurons were entrained to all three rhythms. A similar point holds for the following statements in the Discussion section: "These results suggest that engagement of the interneuron population in all three frequency ranges, and the beta rhythm in particular, may be important for successful information processing" and "Coupled with the fact that interneurons are coherent to beta primarily during correct trials" and "The selective entrainment of the interneuron population to beta during correct trials". These conclusions do not seem to fit the results.

4) In the last paragraph of the Results section, the authors conclude the following: "these results suggest that each rhythmic circuit may differentially contribute task-relevant information". I was confused by this conclusion, considering that the previous sentence states: "each task dimension is equally represented in each rhythmic circuit".

5) In the third paragraph of the Discussion section, the authors write: "Significant spike-phase coherence in the principal cell population was seen primarily during correct trials (Figure 3) […] Beta and low gamma coherent cells exhibited this preference for correct trials more often than theta coherent cells". The differences between beta and low gamma proportions compared to theta proportions seem really small.

6) In the third paragraph of the Discussion section, the authors write: "the majority of principal cells were preferentially coherent to only one rhythm (Figure 3)" (and also the figure legend states: "A large proportion of principal cells exhibit exclusive coherence to one frequency range"). This is clearly true for theta, but many of the gamma coherent cells are also coherent to other rhythms (probably theta). Gamma co-occurs with theta and its amplitude is modulated by theta phase (e.g., Bragin et al., 1995). It seems strange that a significant proportion of cells would be entrained to gamma but not to theta. It might be easier to evaluate these results if the proportions of the total cells entrained to different rhythms were also presented together (e.g., 70% of cells entrained to theta only, 5% entrained to beta only, 5% entrained to gamma only, 5% entrained to gamma and theta, 5% entrained to gamma and beta, 5% entrained to beta and theta, 5% entrained to all three rhythms), as opposed to only presenting the proportions separately for each rhythm class.

7) The "two rhythms" category in Figures 2 and 3 is difficult to evaluate. For example, to which other rhythm are gamma coherent cells entrained? Theta?

8) The following sentence in the fourth paragraph of the Discussion section does not make sense: "Beta coherent interneurons, which were coherent to beta almost exclusively during correct trials, were predominantly also coherent to the other two frequencies (theta and low gamma)".

9) The following sentence in the fourth paragraph of the Discussion section is confusing: "interneurons exhibiting coherence to theta and low gamma irrespective of outcome were more likely to be coherent to all three frequencies during correct trial types than incorrect trial types". "Irrespective of outcome" seems to contradict "more likely to be coherent […] during correct trial types".

10) For the Figure 1 legend, the authors write, "Low gamma amplitude is greater during correct trials than incorrect trials". Yet, in Figure 1, a larger effect is apparent <10 Hz, and this is not mentioned.

*Reviewer #3:*

The authors could improve the paper by addressing the following issues:

1) Figures 2 and 3 distinguish between neurons that phase locked to each rhythm, all three rhythms, and two rhythms. To show the distribution of cells locked to specific pairs (e.g. theta/beta, theta/gamma, etc.) the authors could put this into an additional table. Related to this issue, when neurons fired in phase with more than one rhythm, where the rhythms coordinated? That is, were the cells triggered by power/power, power/phase, or phase/phase comodulation? Finding that different principal cells are best locked to different rhythms raises a fascinating neurocomputational question concerning the mechanisms by which individual and presumably neighboring cells could become phase locked to different rhythms. The authors may want to speculate about this in the Discussion.

2) All of the results are described by proportions of neurons categorized by significant phase locking to in LFP frequency, leaving open the question of whether the magnitude or the specific phase of locking is informative, as suggested by Hasselmo with respect to theta. Finally, the discriminability of phase locked spiking decreases with oscillation frequency, as reflected by the circular histograms showing the distributions in the supplemental figures. To what extent did this influence the reported results?

3) Figure 1 suggests that phase locking across oscillation frequencies varies across the stimulus sampling epoch, with theta locking occurring first, followed by gamma, and then beta. Do the subpopulations of principal cells locked to each of these rhythms also tend to fire at different times from poke onset?

---

## [Author Response]

Reviewer #1:*The study of Rangel et al. investigates the correlation of principal cell and interneuron activity in relation to three neuronal network rhythms emerging in CA1, beta, low gamma and theta rhythms during a context-guided odor-reward association task.*

*Generally, the behavioral and electrophysiological part of the study is perfectly performed. The Introduction and Discussion are very well written. The Results section could be improved in the clarity of the presented results. It is a bit difficult to follow the individual parts of the results, in particular, to keep similarities and differences in the neuronal activity during the various brain rhythms separated. The presentation of the results in Figures 2 and 3 could be improved. The differences between PC and IN recruitment among the various brain rhythms is not easy to understand from the figures. The authors could try to speculate a bit more on the mechanisms which may underlie the emergence of beta activity in the Discussion section.*

We thank the reviewer for the praise with respect to the Introduction and Discussion sections of our study. We also appreciate the reviewer’s feedback regarding the lack of clarity in the presentation of our results. We have edited the Results section to clearly define the comparisons made across specific conditions and have additionally improved the presentation of the data formally in Figures 2 and 3 (now Figures 2 and 5), accordingly. For example, to more thoroughly illustrate differences in the extent to which interneurons and pyramidal cells are engaged in one or multiple rhythms, we have provided supplementary tables ([Supplementary-material SD1-data SD2-data]) and a new figure (Figure 5) showing the proportions of cells that become engaged in every combination of the frequencies analyzed in this study. We hope that these changes better emphasize the meaningful trends in the single cell coherence to each rhythm.

We have also included additional speculation in the Discussion as to the mechanisms that underlie the generation of beta activity in the CA1. Specifically, we cite a new study that utilizes current source density analysis to show that the beta rhythm is likely generated via perforant path input.

Reviewer #2:

*1) The chi-square analyses comparing proportions of cells exhibiting significant spike-phase coherence were not very clearly explained in the Results. For example, in the third paragraph, the authors write: "the majority of beta coherent interneurons exhibited their coherence to beta only during correct trials (chi-square beta = 51.54, d.f., = 2, p < 0.00001)". The chi-square presumably compares the correct category with the incorrect and both categories. How were the expected proportions determined?*

We thank the reviewer for pointing out a specific section in which we could be clearer in describing our comparisons. In this section, all beta coherent cells were assigned to one of three possible categories: those that exhibited significant spike-phase coherence during correct trials only, incorrect trials only, or all trials. To test whether neurons were equally distributed across the three categories, the chi-square compared the proportions found to the expected proportions of 0.33 per category.

If single cell engagement in a rhythm in the form of spike-phase coherence is important for successful processing during the task, one might expect a larger number of cells to be coherent during correct trials only. The converse might be true if single cell spike-phase coherence interferes with successful performance, resulting in a larger number of cells exhibiting significant spike-phase coherence during incorrect trials only. Lastly, the number of cells that exhibit significant spike-phase coherence to a rhythm on all trials (correct and incorrect) might instead be engaged in a processing state that occurs independently of successful task performance. This comparison asks whether the number of cells exhibiting significant spike phase coherence to a rhythm is different across the three categories. We have added additional pairwise comparisons that further describe the differences across the three categories and the extent to which spike-phase coherence to a rhythm is dependent upon performance. We have amended our language in the text to be more explicit about the comparisons presented.

*Similar ambiguities exist for other descriptions of chi-square tests. In the same paragraph, the authors write, "substantial proportions of theta and gamma coherent interneurons exhibited coherence during all trials, irrespective of outcome" and then report significant chi-square statistics for theta and gamma. Did these chi-square tests determine whether the proportions of cells that phase-locked to both correct and incorrect trials was higher than expected? If so, this should be clearly explained.*

While the comparison above tests whether the number of cells in any category was significantly different from the proportion expected across all three categories, the reviewer is correct that it does not directly ask whether the proportion of cells in this category (exhibiting significant spike-phase coherence during all trials) is higher than expected. We have included additional pairwise comparisons to the comparison described above that test whether the differences between any one pair of categories was greater than expected by chance. The addition of these pairwise comparisons allowed us to directly assess whether the proportions in a specific category were higher or lower than expected. We thank the reviewer for pointing out that this more direct test was absent from our study.

*Also, the authors write, "the proportions of interneurons that exhibited coherence contingent on trial outcome differed across the three rhythms" and report significant chi-square tests for theta-beta, theta-gamma, and beta-gamma. What proportions were compared across the different rhythms? What exactly does "contingent on trial outcome" mean? Were correct-selective and incorrect-selective proportions grouped together for these tests? I recommend that the authors re-read all of the chi-square results related to Figures 2 and 3 and make sure that readers can easily determine what was compared.*

The phrase “the proportions of interneurons that exhibited coherence contingent on trial outcome” was meant to describe the interneuron distributions across the three performance categories (now labeled Correct Trials Only, Incorrect Trials Only, or All Trials). We compared the distributions across the three performance categories for all pairs of rhythms in order to determine whether any rhythms are unique in their ability to engage interneuron activity during specific trial types. We have clarified our language in the text, and added post-hoc pairwise comparisons to directly test whether recruitment in one performance category is greater for one rhythm than another.

*2) The authors examine theta, beta, and low frequency gamma. What about high frequency gamma?*

We have added a high gamma frequency range (65-90Hz) to each of the analyses presented in this study.

3) In the third paragraph of the Results section, the authors write: "these results indicate that interneuron entrainment to the beta rhythm is related to successful task performance". I was confused by this conclusion since Figure 2 (middle) shows that more interneurons were entrained to only beta during incorrect trials. For correct trials, Figure 2 (middle) shows that most neurons were entrained to all three rhythms. A similar point holds for the following statements in the Discussion section: "These results suggest that engagement of the interneuron population in all three frequency ranges, and the beta rhythm in particular, may be important for successful information processing" and "Coupled with the fact that interneurons are coherent to beta primarily during correct trials" and "The selective entrainment of the interneuron population to beta during correct trials". These conclusions do not seem to fit the results.

The commentary "these results indicate that interneuron entrainment to the beta rhythm is related to successful task performance" refers to the differences observed when comparing the proportion of cells exhibiting significant spike-phase coherence to Correct Trials only, Incorrect Trials Only, or All Trials (both correct and incorrect). As mentioned in response to one of the reviewer’s previous concerns (see our response to point #1), we have added pairwise comparisons to directly test whether the proportion of cells exhibiting significant spike-phase coherence during correct trials only was significantly greater than expected when compared to any other category. We now refer to this specific result when we make this claim in the Discussion in order to orient readers to the result we believe supports this claim.

We agree with the reviewer that the former Figure 2 (middle) shows that a substantial proportion of cells exhibiting significant spike-phase coherence to beta during incorrect trials only were significantly coherent to a single rhythm. The number of cells that exhibit significant spike-phase coherence to incorrect trials only is a small number, and subsequent pairwise comparisons revealed that the proportion of cells in this group that were coherent to beta only was not significantly different than the proportions observed in correct trials only or all trial categories. In the current version of the manuscript, we have changed the manner in which we demonstrate interneuron and principal cell engagement in multiple rhythms (Figure 2 and Figure 5). This change was made in response to suggestions from all reviewers to clarify the presentation of these results. As a consequence, this comparison is no longer present in the manuscript. We believe that the new presentation of the data, showing the engagement of interneurons and principal cells in all fifteen possible combinations of the four rhythms examined (Figure 5), clarifies these former ambiguities.

4) In the last paragraph of the Results section, the authors conclude the following: "these results suggest that each rhythmic circuit may differentially contribute task-relevant information". I was confused by this conclusion, considering that the previous sentence states: "each task dimension is equally represented in each rhythmic circuit".

In our former presentation of the information analyses, we had included cells that were exclusively coherent to a specific rhythm during Correct Trials Only. In the process of revising our manuscript for resubmission, we adjusted the cell inclusion criteria for many of our analyses (e.g. our magnitude and phase analyses, as well as our new categorization by coherence to combinations of multiple rhythms) to include all cells coherent to a specific rhythm (irrespective of potential coherence to other rhythms) during correct trials (which includes both cells coherent during Correct Trials Only and All Trials). Upon changing these inclusion criteria, we no longer observed the differences in proportion described in our initial version of the manuscript. As a consequence, we no longer believe that the observed differences in the proportion are a robust phenomenon in the data, and have removed it from the manuscript. We instead include a different analysis that examines the extent to which information for a given task dimension changes over the course of the trial in each rhythmic population. We believe that these results represent a more robust phenomenon in the data.

*5) In the third paragraph of the Discussion section, the authors write: "Significant spike-phase coherence in the principal cell population was seen primarily during correct trials (Figure 3) […] Beta and low gamma coherent cells exhibited this preference for correct trials more often than theta coherent cells". The differences between beta and low gamma proportions compared to theta proportions seem really small.*

We agree with the reviewer that this appears to be a very small difference, and one that appears visually negligible. However, since it was a significant difference that we found when we performed a chi-square to examine differences across rhythmic categories, we felt it necessary to report it. We have added commentary to relate this difference to other main findings. We also added a note in the text that this difference is relatively small when compared to the large proportion of theta coherent interneurons that exhibited coherence during All Trials, to orient readers and provide some perspective.

*6) In the third paragraph of the Discussion section, the authors write, "the majority of principal cells were preferentially coherent to only one rhythm (Figure 3)" (and also the figure legend states: "A large proportion of principal cells exhibit exclusive coherence to one frequency range"). This is clearly true for theta, but many of the gamma coherent cells are also coherent to other rhythms (probably theta). Gamma co-occurs with theta and its amplitude is modulated by theta phase (e.g., Bragin et al., 1995). It seems strange that a significant proportion of cells would be entrained to gamma but not to theta. It might be easier to evaluate these results if the proportions of the total cells entrained to different rhythms were also presented together (e.g., 70% of cells entrained to theta only, 5% entrained to beta only, 5% entrained to gamma only, 5% entrained to gamma and theta, 5% entrained to gamma and beta, 5% entrained to beta and theta, 5% entrained to all three rhythms), as opposed to only presenting the proportions separately for each rhythm class.*

We thank the reviewer for suggesting that we present a more detailed breakdown of the proportion of cells coherent to all combinations of the four rhythms examined. We agree that it will improve the reader’s ability to independently interpret this dataset. In this new version of the manuscript, the proportions of cells coherent to one, two, three, or all four rhythms are now presented in [Supplementary-material SD1-data SD2-data]. In the new Figure 5, we now present the number of cells that are coherent to all possible combinations of the four rhythms examined during correct (top) or incorrect (bottom) trials. In this figure, it is clear that the reviewer was correct in predicting that many of the cells exhibiting coherence to two rhythms exhibit coherence to theta. We agree with the reviewer that this is a much clearer way to present the data in the main body of the manuscript.

*7) The "two rhythms" category in Figures 2 and 3 is difficult to evaluate. For example, to which other rhythm are gamma coherent cells entrained? Theta?*

We believe this concern is addressed by our response to the previous question (6).

*8) The following sentence in the fourth paragraph of the Discussion section does not make sense: "Beta coherent interneurons, which were coherent to beta almost exclusively during correct trials, were predominantly also coherent to the other two frequencies (theta and low gamma)".*

We apologize that this line in the text was not clear. The phrase “almost exclusively” refers to the fact that interneuron coherence to beta was overwhelmingly observed primarily during correct trials (Figure 2). However, the interneurons that were coherent to beta during correct trials were often coherent to multiple rhythms. This is now best represented by the large number of interneurons the last category of Figure 5 (top). We have revised the text to clarify that the exclusivity we report refers to coherence to beta during *correct trials*, and not exclusive coherence to the beta rhythm.

*9) The following sentence in the fourth paragraph of the Discussion section is confusing: "interneurons exhibiting coherence to theta and low gamma irrespective of outcome were more likely to be coherent to all three frequencies during correct trial types than incorrect trial types". "Irrespective of outcome" seems to contradict "more likely to be coherent […] during correct trial types".*

We used the phrase “irrespective of outcome” to refer to the cells that were coherent during All Trials (both correct and incorrect trial types) in Figure 2. When we examined whether this population of cells was coherent to one or multiple rhythms, we found that these cells demonstrated different coherence profiles during correct and incorrect types. For example, a cell that was coherent to theta during both correct and incorrect trials could be coherent to all rhythms during correct trials and only theta during incorrect trials. Even though this example cell is coherent to theta during all trials, it is engaged in a different number of rhythmic circuits during correct trials types compared to incorrect trial types. We believe this data is presented in a more transparent manner in the tables provided in [Supplementary-material SD1-data SD2-data], and in a clearer and much simpler manner in Figure 5. We hope that the new figures clarify the results, and we have also revised our language in the text.

*10) For the Figure 1 legend, the authors write, "Low gamma amplitude is greater during correct trials than incorrect trials". Yet, in Figure 1 larger effect is apparent <10 Hz, and this is not mentioned.*

We agree with the reviewer that visually, it does appear to be the case in Figure 1 that there is less theta amplitude during correct pokes than during incorrect pokes. However, in the sentences prior to our reference of this figure, we include statistical tests comparing the mean amplitude of all four frequency ranges during correct and incorrect pokes. There was no significant difference in theta amplitude between correct and incorrect trials.

Reviewer #3:*The authors could improve the paper by addressing the following issues:*

*1) Figures 2 and 3 distinguish between neurons that phase locked to each rhythm, all three rhythms, and two rhythms. To show the distribution of cells locked to specific pairs (e.g. theta/beta, theta/gamma, etc.) the authors could put this into an additional table.*

We thank the reviewer for offering advice as to how to make the presentation of our data clearer to the readers. We have included new tables in [Supplementary-material SD1-data SD2-data] to show the distribution of cells locked to specific combinations of rhythms. In addition, Figure 5 now shows the proportion of cells exhibiting coherence to all possible combinations of the four rhythms examined in this study during correct (top) and incorrect (bottom) trial types. We hope that these changes provide the reader with a much clearer presentation of the results in the main manuscript that is easier to interpret.

*Related to this issue, when neurons fired in phase with more than one rhythm, where the rhythms coordinated? That is, were the cells triggered by power/power, power/phase, or phase/phase comodulation?*

We have attempted to answer this question using two different methods.

a) We performed multi-taper spike-phase coherence analysis and took the magnitude of coherence at four frequencies within each of the four bands used in this study for every correct trial. We then asked whether the magnitude of coherence to one frequency was correlated with the magnitude of coherence to another frequency across trials by performing correlations on the coherence values across trials for every pair of frequencies. We have included the results for this data in Figure 3—figure supplement 1 for the interneuron population and the specific subpopulation of interneurons that are coherent to all four rhythms. Although we are open to the possibility that there is coordinated (e.g. correlated or anticorrelated) coherence to multiple rhythms, there do not appear to be any obvious trends in this dataset.

b) We also attempted to answer this question by looking at local field potential dynamics alone, independently of the spiking activity. For every session, we calculated the instantaneous amplitude of each of the four frequency ranges during correct trials and performed correlations on the amplitude values for every pair of frequencies. We have included the results for this data in Figure 3—figure supplement 1. While there are some pairs of frequencies that appear to be more correlated or anti-correlated than others, all pairs of frequencies appear correlated in some sessions and anti-correlated in others. Thus, we did not find any strong co-modulation or mutual exclusivity for any pair of frequencies in this study.

*Finding that different principal cells are best locked to different rhythms raises a fascinating neurocomputational question concerning the mechanisms by which individual and presumably neighboring cells could become phase locked to different rhythms. The authors may want to speculate about this in the Discussion.*

We agree that this is a very interesting finding in our data. We have added a new paragraph to the Discussion that suggests mechanisms through which neighboring cells could be engaged in different rhythmic circuits and the potential implications of such segregated processing.

*2) All of the results are described by proportions of neurons categorized by significant phase locking to in LFP frequency, leaving open the question of whether the magnitude or the specific phase of locking is informative, as suggested by Hasselmo with respect to theta.*

In response to this comment, we have added a substantial amount of new data to the manuscript (Figure 3, Figure 4, and their supplementary figures) that examines changes in the magnitude and phase of coherence across correct and incorrect trials. The magnitude and phase of coherence were additionally informative measures that complemented the data described in proportions. We sincerely thank the reviewer for this suggestion, as we believe these additions greatly strengthen the claims in our original manuscript.

*Finally, the discriminability of phase locked spiking decreases with oscillation frequency, as reflected by the circular histograms showing the distributions in the supplemental figures. To what extent did this influence the reported results?*

We thank the reviewer for bringing this concern to our attention. We believe that this could be reflected in the large number of interneurons and principal cells exhibiting significant spike-phase coherence to theta. Similarly, the magnitudes of coherence to theta were often greater than the magnitudes observed with respect to the other three rhythms. However, we did not observe any significant differences in the magnitudes of coherence to beta, low gamma, or high gamma, indicating that the observed differences across these three frequency ranges are not related to a biased ability to discriminate phase locking in lower frequencies.

*3) Figure 1 suggests that phase locking across oscillation frequencies varies across the stimulus sampling epoch, with theta locking occurring first, followed by gamma, and then beta. Do the subpopulations of principal cells locked to each of these rhythms also tend to fire at different times from poke onset?*

In response to this question, we calculated the mean firing rate during 250ms time bins within correct trial nose pokes for all interneurons and principal cells exhibiting coherence to a specific rhythm. The normalized mean firing rates for each population are shown in Figure 7. We did not observe any consistent relationships between mean firing rates and mean amplitude changes in each rhythm (Figure 1). We have thus chosen not to include this data in the current version of the manuscript.

Author response image 1.**DOI:**
http://dx.doi.org/10.7554/eLife.09849.021